



# First report of ultra-high pressure metamorphism in the Paleozoic Dunhuang orogenic belt (NW China): Constrains from *P-T* paths of garnet clinopyroxenite and SIMS U-Pb dating of titanite

Zhen M.G. Li [1], Hao Y.C. Wang [1], Qian W.L. Zhang [1], Meng-Yan Shi [1], Jun-Sheng Lu [2],

Jia-Hui Liu [1], and Chun-Ming Wu [1*]

[1] College of Earth and Planetary Sciences, University of Chinese Academy of Sciences,

P.O. Box 4588, Beijing 100049, China

[2] State Key Laboratory of Continental Dynamics, Department of Geology, Northwest

University, Xi'an 710069, China

* Corresponding author. Tel: +86 10 8825 6312; fax: +86 10 8825 6012; E-mail: wucm@ucas.ac.cn





## Abstract

Ultra-high pressure (UHP) metamorphism is recorded by garnet clinopyroxenite
enclaves enclosed in an undeformed, unmetamorphosed granitic pluton, northeastern
Paleozoic Dunhuang orogenic belt, northwest China. Three to four stages of metamorphic
mineral assemblages have been found in the garnet clinopyroxenite, and clockwise
metamorphic pressure-temperature (*P-T*) paths were retrieved, indicative of metamorphism
of a possible subduction environment. Peak metamorphic *P-T* conditions (790~920 ℃ /
28~41 kbar) of garnet clinopyroxenite suggest that they experienced high pressure to UHP
metamorphism, and the UHP metamorphism occurred in the coesite- or diamond-stability
field. The UHP metamorphic event is further confirmed by the occurrence of high-Al
titanite enclosed in the garnet, along with at least three groups of aligned rutile lamellae
exsolved from within the garnet. SIMS U-Pb dating of metamorphic titanite indicates that
the post peak, subsequent tectonic exhumation of the UHP rocks occurred in the Devonian
(~389~370 Ma). These data suggest that part of the Paleozoic Dunhuang orogenic belt
experienced UHP metamorphism, and diverse metamorphic facies series prevailed in this
orogen in the Paleozoic. It can be further inferred that most of the UHP rocks of this orogen
are now buried in the depth.









## Introduction


It is well known that ultra-high pressure (UHP) metamorphism refers
to metamorphic pressure high enough to stabilize coesite, i.e., pressure reaches at least ~2.7
GPa if temperature reaches ~700 ℃. The UHP metamorphism is anticipated to be formed
in the subduction process at very low thermal gradient of usually less than 10 ℃ / km, or
even as low as ~5 ℃ / km. In orogenic belts, UHP metamorphism can be validly certified
by presence of diagnostic minerals such as coesite (e.g., Chopin, 1984; Smith, 1984),
diamond (e.g., Sobolev and Shatsky, 1990; Xu et al., 1992), Na-Ti-P-bearing garnet (e.g.,
Ye et al., 2000) or even pseudomorph of stishovite (e.g., Liu et al., 2007, 2018). But
unfortunately, in some orogenic belts these diagnostic minerals cannot be found, which in
turn, brings people some uncertainties in recognizing UHP metamorphism. In this
contribution, we present UHP metamorphism recorded in garnet clinopyroxenite enclaves
within an undeformed, unmetamorphosed granitic pluton, northeast Paleozoic Dunhuang
orogenic belt, northwest China.
The Dunhuang area has long been considered as an ancient stable block formed in the
Precambrian. Until recently, clockwise metamorphic *P-T* paths of eclogite, mafic granulite,
amphibolite and metapelite, typical metamorphic products of subduction background, were
retrieved elsewhere in this region (Zhang et al., 2012; Zong et al., 2012; He et al., 2014;
Peng et al., 2014; Zhao et al., 2016; Wang et al., 2016, 2017a, b, 2018a, b; Zhang et al.,
2020). The metamorphic event was dated to have occurred in the Silurian to Devonian era
(Zong et al., 2012; He et al., 2014; Wang et al., 2016, 2017a, b, 2018a, b; Zhang et al.,
2020). These enabled people to believe that this area was a Paleozoic orogenic belt (Zhao
et al., 2016; Wang et al., 2017a, b; 2018a, b; Zhang et al., 2020), albeit the subduction



polarity remains ambiguous. Eclogite (Wang et al., 2017a) and high-pressure mafic
granulite (Zong et al., 2012; He et al., 2014; Wang et al., 2016, 2017a, b, 2018b; Zhang et
al., 2020) have been found in this orogen, but UHP rocks have not been discovered before,
which in turn, limits our understanding of the orogenic process as a whole.

## Regional geology

The Dunhuang orogenic belt strikes SWW-NEE and covers an area of approximately

440 km long and 100 km wide. It is tectonically bordered by the Paleozoic Beishan
orogenic belt to the north, the Precambrian Tarim craton to the west, the Precambrian Alexa
block to the east, and the Paleozoic Altyn Tagh-Qilian orogenic belt to the south (Fig. 1).
The Dunhuang orogenic belt was dismembered by sinistral strike-slip faults to several
tectonic blocks (Fig. 2), possibly in the Tertiary. The prominent characteristics of the
Dunhuang orogenic belt is that at least in the Hongliuxia, Qingshigou, Kalatashitage and
Mogutai-Dongbatu blocks, eclogite, high- and medium-pressure mafic granulite, and
amphibolite occur as rootless tectonic lenses or puddings enclosed within the metapelite
and metasandstone matrix (Wang et al., 2016, 2017a, 2018a, b), indicative of typical block-
in-matrix feature of tectonic mélange (Festa et al., 2012). Some closely amalgamated
tectonic-metamorphic slices can also be found in northwest Dunhuang orogenic belt
(Zhang et al., 2020), which were metamorphosed in obviously different depths and were
later juxtaposed in the same crustal level in the tectonic exhumation.

Unfortunately, coesite or diamond, either as inclusion or inter-granular minerals, have

not been found from the eclogite, mafic granulite, amphibolite or metapelite, therefore,
UHP metamorphism of this orogen has not been found before. Recently, we found high-Al

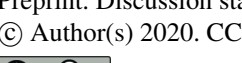



titanite-bearing garnet clinopyroxenite and obtained UHP *P-T* conditions of such rocks in
the Daquan area, northeast Dunhuang orogenic belt (Fig. 3). Garnet clinopyroxenite occurs
as enclaves enclosed in an undeformed, unmetamorphosed granite body (Figs. 3B, 4A-B),
but crystallization age of the granite cannot be determined due to severe decrystallization
of magmatic zircon caused by radioactive damage.

**Petrography**


Retrograde symplectite rimming the embayed, relict garnet of the garnet
clinopyroxenite can be easily seen in the outcrop (Fig. 4C). Micropetrographic features of
the garnet clinopyroxenite are depicted in Figures 5 and 6. Three to four stages of
metamorphic mineral assemblages were found in the four representative samples. The
mineral abbreviations are from Whitney and Evans (2010) hereafter, and subscripts of the
minerals 1, 2, 3, and 4 refer to the corresponding minerals formed at the sequential four
metamorphic stages, respectively, throughout this paper.
All the four samples are mainly bimineralic, consisting of garnet and clinopyroxene.
The prograde assemblage (M1) is represented by the fine-grained ilmenite ($Ilm_1$) +
hornblende ($Hbl_1$) + plagioclase ($Pl_1$) $\pm$ clinopyroxene ($Cpx_1$) inclusions enclosed in the
garnet ($Grt_2$) (Figs. 5A, B, C). High-Al titanite ($Ttn_1$) also appears as inclusion within
garnet (Figs. 6A-B). The peak metamorphic assemblage (M2) consists of garnet ($Grt_2$) and
clinopyroxene ($Cpx_2$), plus matrix minerals including minor rutile ($Rt_2$), magnetite ($Mag_2$),
high-Al titanite ($Ttn_2$) and apatite ($Ap_2$), as shown in Figures. 5A-C. The matrix rutile ($Rt_2$)
is rare (Fig. 5G). The retrograde assemblage (M3) is mainly the warm-like symplectite,
consisting of fine-grained plagioclase ($Pl_3$), hornblende ($Hbl_3$) and ilmenite ($Ilm_3$)





intergrowth, riming the garnet (Figs. 5A-C, H). Similar decomposition textures in mafic
granulites can be found elsewhere and are repeatedly demonstrated to be formed under
severe decompression during tectonic exhumation (e.g., Wang et al., 2016, 2017a, b;
Petrakakis et al., 2018; Zhang et al., 2020). Other retrograde assemblages include fine-
grained ilmenite ($Ilm_3$) ± plagioclase ($Pl_3$) ± hornblende ($Hbl_3$) ± titanite ($Ttn_3$) lamellae
exsolved from within the clinopyroxene ($Cpx_2$) (Fig. 5I), hornblende ($Hbl_3$) retrograded
from clinopyroxene ($Cpx_2$) (Fig. 5F), ilmenite ($Ilm_3$) retrograded from rutile ($Rt_2$) (Fig. 5G),
and ilmenite ($Ilm_3$) retrograded from high-Al titanite ($Ttn_2$) (Fig. 6B), as well as aligned
rutile lamellae ($Rt_3$) exsolved from within the garnet in three different directions (Figs. 6C-
D). Occasionally, the final retrograde assemblage ($M_4$) can be found, i.e., actinolite ($Act_4$)
and chlorite ($Chl_4$) retrograded from hornblende ($Hbl_3$) (Fig. 5H). In sample 17D95,
especially, minor spinel ($Spl_3$) can be found, and it mainly coexists with tremolite ($Tr_3$) ±
rutile ($Rt_3$) as idiomorphic retrograded phases within the garnet (Fig. 6E), possibly
exsolved from within garnet, similar to the clinopyroxene exsolved from within the garnet
in eclogite in Sulu orogenic belt, eastern China (Ye et al., 2000). Such inclusion-like
minerals were in fact decomposed from garnet (Hwang et al., 2019). In sample 17D80,
there is idiomorphic hexagon ilmenite ($Ilm_3$) in garnet, and separated by high-Al titanite
($Ttn_2$) from midcourt line (Fig. 6F), and such reaction textures are similar to those in
granulite facies metapelite and may represent extremely high *P* or *T* conditions (e.g., Ague
and Eckert, 2012).

**Metamorphic *P-T* paths**
***Mineral chemistry***



Compositional analyses, backscattered electron (BSE) images, as well as X-ray
compositional mapping of minerals were determined by electron probe microanalysis
(EPMA) using a JOEL JXA-8230 analyzer at the School of Resource and Environmental
Engineering, Hefei University of Technology, China. The  analytical conditions were 15
kV accelerating voltage and 20 nA beam current and the counting time was 10–20 s.
Usually, 3~5 μm electron beam diameter was used, while 3 μm electron beam size was
only adopted in analyzing the tiny minerals. Natural minerals were used as standards, and
the ZAF program was utilized for matrix corrections. Generally, at least 3~5 grains were
analyzed for any representative mineral, and 1~60 spots of each grain were probed. The
representative mineral compositions are listed in Table S1 and the computed *P-T* conditions
are listed in Table 1. Ferric iron content of both clinopyroxene and garnet was determined
by stoichiometric and charge balance criteria (Droop, 1987), while ferric iron content of
hornblende was evaluated by the method of Holland and Blundy (1994).
The garnet (Grt$_2$) is chemically homogeneous in each sample and is mainly consisting
of almandine ($X_{Alm}$=0.34~0.54), pyrope ($X_{Prp}$=0.19~0.48) and grossular ($X_{Grs}$=0.18~0.32)
but negligible spessartine components. Such garnet is chemically different from those in
mantle xenolith or eclogite. Negligible chemical zonation of the garnet was found. In the
very rim of the garnet, the Fe# [=Fe/(Fe+Mg)] value increases slightly (Table S2; Figs. 7,
8), indicative of post-peak Fe-Mg diffusion between the garnet rim and adjacent
clinopyroxene and / or decomposition of the garnet rim (Spear and Florence, 1992). This
is also demonstrated by micropetrography (Figs. 5 and 6). Chemical analytical profiles
(Table S3) suggest that the clinopyroxene (Cpx$_2$) is almost chemically homogeneous in
each sample and is essentially diopside based on the classification of Morimoto (1988) (Fig.

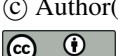



S1) with negligible jadeite fraction. Due to exsolution, however, chemical composition of
most of the $Cpx_2$ was altered to different extent. Although the $Mg^{2+}$ and $Fe^{2+}$ cations of the
$Cpx_2$ grains are generally homogeneous, but $Al^{3+}$ and $Ca^{2+}$ cations show somewhat
variations (Figs. S2, S3, S4, S5), thus the reintegrated chemical composition of
clinopyroxene was used to estimate peak *P-T* conditions. High-Al titanite contains
remarkable $Al_2O_3$ (8.2~10.2 wt%) and F contents (1.3~2.8 wt%), which signify HP / UHP
pressure metamorphism (e.g., Smith, 1981; Franz and Spear, 1985).
*Geothermobarometry*
Metamorphic *P-T* conditions of the peak metamorphism (M2) were determined by the
garnet-clinopyroxene geothermometer (Nakamura, 2009) coupled with the garnet-
clinopyroxene geobarometer (Beyer et al., 2015), using averaged chemical composition of
garnet and reintegrated chemical composition of clinopyroxene. Although this
geobarometer was experimentally calibrated for mantle eclogite, however, chemical
compositions of the natural rocks reported in this work are similar to those of the
experimental run products (Beyer et al., 2015). Accuracy of this geobarometer is estimated
to be ±4 kbar (Beyer et al., 2015). The prograde (M1) and retrograde (M3) assemblages are
mainly consisting of plagioclase and hornblende but without quartz, therefore, *P-T*
conditions of the M1 and M3 assemblages were estimated by the monomineralogic
hornblende geothermobarometers (Gerya et al., 1997).
*Metamorphic P-T paths*
Metamorphic *P-T* path of sample 17D78 passes from 662 ℃ / 5.4 kbar (M1) through
789 ℃ / 28 kbar (M2) to 621 ℃ / 4.6 kbar (M3). As for the other three samples,
metamorphic *P-T* paths were estimated respectively as the follows: sample 17D80, 902 ℃





/ 38.2 kbar (M2) → 656 °C / 5.4 kbar (M3); sample 17D90, 695 °C / 7.2 kbar (M1) → 868°C
/ 31.8 kbar (M2) → 669 °C / 6.0 kbar (M3); sample 17D95, 918°C / 41.3 kbar (M2) →
631 °C / 5.6 kbar (M3). The retrieved metamorphic *P-T* paths of the garnet clinopyroxenite
enclaves are all clockwise (Fig. 9), indicative of subduction zone setting (c.f., Ernst, 1988;
Harley, 1989). It should be stated that the peak metamorphism (except for sample 17D78)
lies in the coesite or diamond stability field (Fig. 9), certifying UHP metamorphism. The
UHP conditions is further evidenced by the occurrence of at least three groups of aligned
rutile lamellae ($Rt_3$) exsolved from within the garnet (Figs. 6C-D) and chemically
homogeneous high-Al titanite ($Ttn_2$) enclosed in the garnet (Figs. 6A-B), being
characterized by $X_{Al}$ [$=Al/(Al+Fe^{3+}+Ti)$]=0.25~0.29. These two mineralogical
characteristics together indicate UHP metamorphism (c.f., Ye and Ye, 1996; Tropper et al.,
2002; Ague and Eckert, 2012).

**Dating metamorphism**

No zircon was found in these samples, possibly due to $SiO_2$-undersaturated bulk
composition of these rocks. Therefore, SIMS U-Th-Pb dating of metamorphic titanite
scenario was chosen to determine the age of metamorphism (in this case, the cooling age).
The SIMS U-Th-Pb analyses of titanite were performed using a Cameca IMS-1280HR
SIMS at Institute of Geology and Geophysics, Chinese Academy of Sciences, Beijing,
China. The instrument description and analytical procedure for titanite dating is identical
to that of dating perovskite (Li et al., 2010) and has been described in detail in Li et al.
(2014) and Ling et al. (2015), thus only a brief summary is described here. The $O_2^-$ primary
ion beam was accelerated at ~13 kV, with an intensity of ~9 nA. The ellipsoidal spot is



about 20 μm × 30 μm in size. The $^{40}Ca^{48}Ti_2^{16}O_4^+$ peak is used as a reference peak for
centering the secondary ion beam, energy and mass adjustments. A mass resolution of
~7000 (defined at 50% peak height) was used. A single electron multiplier was used in ion-
counting mode to measure secondary-ion beam intensities by a peak jumping sequence,
including isotopes of $Pb^+$, $Th^+$, $U^+$, $ThO^+$, $UO^+$, and $^{40}Ca^{48}Ti_2^{16}O_4^+$ to produce one set of
data. Analyses of the standard YQ82 titanite were interspersed with unknown grains. Each
measurement consists of 15 cycles, and the total analytical time is ~19 min. Pb/U
calibration was performed relative to YQ82 titanite standard ($^{206}Pb/^{238}U$ age = 1837.6 Ma,
Li et al., 2016). U and Th concentrations were calibrated against titanite BLR-1 (Aleinikoff,
et al., 2007). A long-term uncertainty of 1.5% (1σ RSD) for $^{206}Pb/^{238}U$ measurements of
the standard titanite was propagated to the unknowns, despite that the measured $^{206}Pb/^{238}U$
error in a specific session is generally ≤ 1% (1σ RSD). A Tera-Wasserburg (Tera and
Wasserburg, 1972) plot was constructed with common lead uncorrected data to deduce the
common lead composition, then a $^{207}Pb$-based common lead correction method was
conducted to single analysis. Data reduction was carried out using the Isoplot/Ex v. 2.49
program (Ludwig, 2001). Uncertainties on individual analysis in data tables are reported at
1σ level. The final U-Pb age result is quoted with 95% confidence interval (Table S4).
The BSE images of titanite are shown in Figure S6. The titanite images are
homogeneous in sample 17D78, while in samples 17D90 and 17D95 they are altered to
different extent. The resulted U-Pb ages of the metamorphic titanite from samples 17D78
and 17D95 are ~370±9 Ma and ~389±8 Ma (Table S4; Fig. 10), respectively, while sample
17D95 records a younger age of ~362±7 Ma. However, the ages of titanite of sample 17D90
are scattered and younger, and have an age spectrum peak of ~253±14 Ma. When





considering that the U-Pb closure temperature of titanite is about 660~700 ℃ (Scott and
St-Onge, 1995) or 750~790 ℃ (Sun et al., 2012), the U-Pb age of the titanite (~389-370
Ma) possibly records the cooling / retrograde period after the peak UHP metamorphic event,
i.e., timing of (earlier) tectonic exhumation.

**Discussion**

Although no coesite or diamond was found in the Dunhuang orogenic belt, UHP

metamorphism of the orogen is evidenced by the garnet clinopyroxenite enclaves in a
limited location. However, problems concerning rock type, micropetrography and *P-T*
computation should be discussed in detail, in order to trustfully demonstrate the UHP
metamorphism.
***About the rock type***

It is well known that skarn or calcsilicate always consists of Ca-rich minerals

including garnet, diopside, wollastonite, scapolite, vesuvianite, calcite, quartz, and the
garnet is mainly consisting of andradite and grossular components (e.g., Ryan-Davis et al.,
2019; Alaminia et al., 2020). However, the garnet clinopyroxenite reported in this work is
obviously not skarn or calcsilicate, because both the mineral assemblages and chemical
composition of the garnet undoubtedly do not match that of skarn or calcsilicate.

The clinopyroxene of this work is essentially $Na_2O$-$Al_2O_3$-deficient ($Na_2O$ < 0.35

wt%, $Al_2O_3$ < 3.0 wt%) and the jadeite phase component of clinopyroxene is negligible.
Therefore, although chemical compositions of the garnet and clinopyroxene somewhat
overlap that of mantle xenolith and eclogite, the present garnet clinopyroxenite is neither
mantle xenolith nor mantle eclogite or crustal eclogite. Furthermore, the prograde





assemblage (M1) clearly indicates the subduction process. It is therefore suggested that the
protolith of the garnet clinopyroxenite might be subducted to very deep but different depths
and thus record clockwise *P-T* paths (Fig. 9).
***UHP metamorphism evidenced by reaction textures***

The rocks contain high-Al titanite enclosed in the garnet and preserve three groups of

aligned rutile lamellae exsolved from within the garnet (Fig. 6). But, it is noted that high-
Al titanite also appears in low-*P* metamorphic rocks (e.g., Enami et al., 1993; Castelli and
Rubatto, 2002), and the activity of F and bulk-rock composition also affect the Al and F
contents of titanite (Franz and Spear, 1985; Enami et al., 1993; Carswell et al., 1996).
However, the low-*P* rocks they reported are essentially skarn, which contains considerable
calcite, and contains negligible pyrope in garnet. Furthermore, there are no aligned rutile
lamellae in the garnet of their skarn. Castelli and Rubatto (2002) suggest that if there are
appropriate bulk compositions with high fluorine activities, high-Al titanite could also be
formed at high-*T* rather than high-*P* conditions. However, their modeling is based on the
carbonate ($CaO-TiO_2-SiO_2-H_2O-CO_2$) system, quite different from our samples. In
addition, the An-rich plagioclase can impede the stabilization of high-Al titanite (Oberti et
al., 1991), thus the occurrence of anorthite in prograde assemblages indicates that high-Al
titanite formed during peak metamorphism, coexisting with garnet.

However, except for UHP metamorphism, the rutile lamellae exsolved from within

the garnet could also be formed at high-*T* conditions (>900 ℃, especially in high-*P*
granulite) (e.g., Snoeyenbos et al., 1995), actually the Al content of titanite increases with
*P* and decreases with *T* (Smith, 1980, 1981, 1988). In this regard, in spite of the peak high-
*T* condition, the effect of *P* should still play a major role.





### *UHP metamorphism confirmed by valid geothermobarometers*

For estimating metamorphic *P-T* conditions of garnet clinopyroxenite, the garnet-clinopyroxene geothermometer and the garnet-clinopyroxene geobarometer are quite necessary and are in fact irreplaceable. As we know, at least 30 versions of the garnet-clinopyroxene Fe-Mg exchange geothermometer have been calibrated in the past five decades. The most recent Nakamura (2009) thermometer was calibrated based on data collected from the literature of phase equilibrium experiments in mafic and ultramafic systems, and the standard error is relatively small (±74 ℃) in reproducing all the available experimental data, in the experimental *P-T* ranges 800~1820 ℃ / 15~75 kbar (Nakamura, 2009). On the contrary, previous formulations of the garnet-clinopyroxene geothermometer are inconsistent with the compiled experimental data set, and they either underestimate *T*s by about 100 ℃ when *T* >1300 ℃ or overestimate *T*s by 100~200 ℃ when *T* <1300 ℃ (Nakamura, 2009). Furthermore, former garnet-clinopyroxene geothermometers tend to overestimate *T*s for high-Ca garnet (Xgrs = 0.30~0.50), as found by Nakamura (2009). Therefore, because of its wide representative and relatively high accuracy, the Nakamura (2009) geothermometer was adopted in this paper. It should be stated that grossular component of the garnet ranges between 0.17~0.32, and chemical compositions of garnet and clinopyroxene fall within the calibration range of this geothermometer, therefore, certifies its applicability in these samples.

As for estimating metamorphic *P*s of the samples, except for one garnet-clinopyroxene geobarometer calibrated based on Ca-Mg exchange between garnet and clinopyroxene (Brey et al., 1986) in the CMAS system for magnesian garnet (Xpyr > 0.8), all the other garnet-clinopyroxene geobarometers (Mukhopadhyay, 1991; Simakov and





Taylor, 2000; Simakov, 2008; Beyer et al., 2015) were calibrated based on net-transfer
model reactions between garnet and clinopyroxene, involving grossular and pyrope
components in garnet and diopside, as well as Ca-tschermak and enstatite components in
clinopyroxene. These garnet-clinopyroxene geobarometers are made in the mafic or
ultramafic system, and are applicable to mantle eclogite with high-Na clinopyroxene, or
garnet clinopyroxenite with low-Na clinopyroxene which is the case for rocks reported in
this work (Na in Cpx < 0.02, based on 6 O basis). In the computation, we don't need
activities of either jadeite or acmite phase components in clinopyroxene and therefore,
errors of chemical compositions of low-Na clinopyroxene do not translate to larger pressure
errors in applying these geobarometers. Among different versions of the garnet-
clinopyroxene geobarometers, the Beyer et al. (2015) barometer was calibrated based on
phase equilibrium experimental data in the *P-T* ranges of 2~7 GPa / 900~1550 ℃. Standard
error of this barometer is approximately ±4 kbar (Beyer et al., 2015), and this barometer
was applied to our samples because it is the most accurate version and chemical
compositions of our mineral samples are similar to the experimental run products.
It should be stated that for our garnet and clinopyroxene, the $Cr_2O_3$ components are
negligible (garnet, 0.02~0.04 wt%; clinopyroxene, 0.02~0.04 wt%), therefore, the $Cr_2O_3$-
based garnet-clinopyroxene geobarometers (Mercier, 1980; Taylor and Nimis, 1998; Nimis
and Taylor, 2000) cannot be applied.
The yielded *P-T* conditions of peak metamorphism lie in the UHP metamorphic region
for samples 17D95, 17D90 and 17D80, or high-*P* region for sample 17D78 which lies
slightly lower than the coesite-quartz transition curve (Fig. 9). However, when considering





error (±4 kbar) of the garnet-clinopyroxene geobarometer (Beyer et al., 2015), it can be
concluded that peak metamorphism occurred at UHP conditions for all the samples.
***Possible fast exhumation of the UHP metamorphic rocks***

It is generally believed that UHP rocks should experience fast tectonic exhumation

from the depth, otherwise the UHP assemblages may be replaced by lower-*P* assemblages.
Thus, when considering the closure temperatures (660~700 ℃, Scott and St-Onge, 1995;
or 750~790 ℃, Sun et al., 2012) of the U-Pb system of titanite which are slightly lower
than the peak metamorphic temperatures (790~920 ℃), SIMS U-Pb dating of metamorphic
titanite possibly records the ages (~389-370 Ma) of tectonic exhumation, postdating but
very approaching the peak metamorphism. In fact, the metamorphic *P-T* paths of the garnet
clinopyroxenite (Fig. 9) also suggest relatively rapid uplift, albeit the retrograde *P-T* paths
are hybrid of the western Alpine and Franciscan types (Ernst, 1988). Furthermore, large
gaps of the peak metamorphic pressures among these samples suggest that these rocks were
subducted to different depths and were later amalgamated at the same crustal level during
tectonic exhumation.

Although the UHP rocks were found within a limited granitic pluton, however, it is

reasonable to infer that other UHP rocks, either more or less, are buried in the root of the
Dunhuang orogenic belt. Furthermore, it is found both medium and high *P/T* facies series
metamorphism occurred in this orogen in the Silurian to Devonian (Wang et al., 2017a, b,
2018a, b; Zhang et al., 2020). The UHP garnet clinopyroxenite reported in this contribution
further demonstrates that different metamorphic *P/T* facies series may prevail in a same
orogenic belt, which is neglected to different extent in the past.



## Conclusion


Three to four stages of metamorphic mineral assemblages are preserved in the garnet
clinopyroxenite enclaves within a granitic pluton in the northeast Paleozoic Dunhuang
orogenic belt, northwest China. The peak metamorphism (870~920 ℃ / 32~41 kbar)
occurred in the coesite or even the diamond stability field, and the concurrence of the high-
Al titanite and at least three groups of aligned rutile lamellae exsolved from within the
garnet further confirm the UHP metamorphic event. Clockwise metamorphic *P-T* paths of
the garnet clinopyroxenite were retrieved, indicative of subduction process. SIMS U-Pb
dating of metamorphic titanite indicates that the tectonic exhumation of the ultra-high
pressure metamorphic rocks might occurred in the Devonian (~389-370 Ma), postdating
the peak UHP metamorphism. It should be noted that both medium- and high-*P/T* facies
series metamorphism occurred in this Paleozoic orogen. Furthermore, it is reasonable to
infer that most of the UHP rocks are buried in depth, possibly in the root of this orogen.

*Data availability*. The data set is given in Supplement.

*Supplements*. The supplement related to this article is available online at:

**Figure S1.** Classification of clinopyroxene in different samples (classification of
Morimoto, 1988).
**Figure S2.** EPMA analytical transverses of clinopyroxene (sample 17D78).
**Figure S3.** EPMA analytical transverses of clinopyroxene (sample 17D80).

**Figure S4.** EPMA analytical transverses of clinopyroxene (sample 17D90).
**Figure S5.** EPMA analytical transverses of clinopyroxene (sample 17D95).



**Figure S6.** Backscattered electron images of titanite separated from garnet clinopyroxenite
samples for SIMS U-Pb dating. (a) Sample 17D78. (b) Sample 17D95. (c) Sample 17D90.
The circles with red figures represent analytical spots. The yellow numbers are the
respective $^{207}$Pb-based common lead corrected ages involved in the calculation for samples
17D78 and 17D90, while the white and yellow numbers are the respective 207Pb-based
common lead corrected ages both involved in the calculation for sample 17D95, and
obtained the younger and older mean ages, respectively.
**Table S1.** Chemical compositions of the representative minerals.
**Table S2.** Chemical compositional profiles of the garnet.
**Table S3.** Chemical compositional profiles of the clinopyroxene.
**Table S4.** SIMS U-Th-Pb analytical data for titanite separated from garnet clinopyroxenite.

*Author contributions*. Chun-Ming Wu and Hao Y.C. Wang guided the field work, they
and Zhen M.G. Li, Qian W.L. Zhang and Meng-Yan Shi did the field investigation. Zhen
M.G. Li, Qian W.L. Zhang and Jia-Hui Liu carried all the experiments. Zhen M.G. Li, Jun-
Sheng Lu, Qian W.L. Zhang and Jia-Hui Liu processed the data. Zhen M.G. Li drew all
the figures. Zhen M.G. Li and Chun-Ming Wu wrote the original manuscript and all the
authors revised the manuscript and proved submission of it.

*Competing interests*. The authors declare that they have no conflict of interest.

*Acknowledgements*. Professor Yonghong Shi and Dr. Juan Wang guided the authors in
electronic microprobe analysis. We thank Professor Qiu-Li Li and Dr. Xiao-Xiao Ling for



their assistance in SIMS U-Pb dating of titanite at Institute of Geology and Geophysics,
Chinese Academy of Sciences, China.

*Financial support*. This work was supported by the National Natural Science Foundation
of China (41730215) and the Key Research Program of Frontier Sciences from the Chinese
Academy of Sciences (QYZDJ-SSW-DQC036).

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

Preliminary report on the metamorphic evolution of the Guanyingou amphibolites,

Dunhuang Metamorphic Complex, NW China. Acta. Petrol. Sin., 30, 503–511, 2014.

[in Chinese with English abstract]

Petrakakis, K., Schuster-Bourgin, N., Habler, G., and Abart, R.: Ca-rich garnets and

associated symplectites in mafic peraluminous granulites from the Gföhl Nappe

System, Austria, Solid Earth, 9, 797–819, https://doi.org/10.5194/se-9-797-2018,

2018.

Ryan-Davis, J., Lackey, J. S., Gevedon, M., Barnes, J. D., Lee4, C-T. A., Kitajima, K., and

Valley, J. W.: Andradite skarn garnet records of exceptionally low $\delta^{18}O$ values within





an Early Cretaceous hydrothermal system, Sierra Nevada, CA, Contrib. Mineral. Petr.,
174, 68, https://doi.org/10.1007/s00410-019-1602-6, 2019.
Scott, D. J., and St-Onge, M. R.: Constraints on Pb closure temperature in titanite based on
rocks from the Ungava orogen, Canada: Implications for U-Pb geochronology and P-
T-t path determinations, Geology, 23, 1123-1126, https://doi.org/10.1130/0091-
7613(1995)023<1123:copcti>2.3.co;2, 1995.
Simakov, S. K., and Taylor, L.A.: Geobarometry for mantle eclogites: solubility of Ca-
tschermaks in clinopyroxene, Int. Geol. Rev., 4, 534–544,
https://doi.org/10.1080/00206810009465097, 2000.
Simakov, S. K.: Garnet-clinopyroxene and clinopyroxene geothermobarometry of deep
mantle and crust eclogites and peridotites, Lithos, 106, 125–136,
https://doi.org/10.1016/j.lithos.2008.06.013, 2008.
Smith, D. C.: Coesite in clinopyroxene in the Caledonides and its implications for
geodynamics, Nature, 310, 641-644, https://doi.org/10.1038/310641a0, 1984.
Smith, D.C.: A review of the peculiar mineralogy of the "Norwegian coesite eclogite
province", with crystal-chemical, petrological, geochemical and geodynamical notes
and an extensive bibliography, In: Smith, D.C. ed. Developments and Petrology 12,
Eclogite-facies Rocks. Amsterdam: Elsevier, 1988.
Smith, D.C.: Highly aluminous sphene (titanite) in natural high-pressure hydrous-eclogite-
facies rocks from Norway and Italy, and in experimental runs at high pressure, 26th
International Geological Congress, Paris, France (Abstract), Section 02.3.1, 1980.





Smith, D.C.: The pressure and temperature dependence of Al-solubility in sphene in the

system Ti-Al-Ca-Si-O-F, Progress Experimental Petrology, Series D 18, 193-197,

1981.

Snoeyenbos, D.R., Williams, M.L., and Hanmer, S.: Archean high-pressure metamorphism

in the western Canadian Shield. Eur. J. Mineral., 7, 1251–1272,

https://doi.org/10.1127/ejm/7/6/1251, 1995.

Sobolev, N. V., and Shatsky, V. S.: Diamond inclusions in garnets from metamorphic rocks:

a new environment for diamond formation, Nature, 343, 742-746,

https://doi.org/10.1038/343742a0, 1990.

Spear, F. S., and Florence, F. P.: Thermobarometry in granulites: pitfalls and new

approaches, Precambrian. Res., 55, 209-241, https://doi.org/10.1016/0301-

9268(92)90025-j, 1992.

Spear, F.S.: Metamorphic Phase Equilibria and Pressure-Temperature-Time Paths.

Washington, D.C., Mineralogical Society of America, 1993.

Sun, J., Yang, J., Wu, F., Xie, L., Yang, Y., Liu, Z., and Li, X.: In situ U-Pb dating of

titanite by LA-ICPMS, Chinese. Sci. Bull., 57, 2506-2516,

https://doi.org/10.1007/s11434-012-5177-0, 2012.

Taylor, W. R., and Nimis, P.: A single-pyroxene thermobarometer for lherzolitic Cr-

diopside and its application in diamond exploration. Seventh International Kimberlite

Conference Abstract Volume, Cape Town, pp. 897–898, 1998.

Tera, F., and Wasserburg, G. J.: U-Th-Pb systematics in three Apollo 14 basalts and the

problem of initial Pb in lunar rocks, Earth. Planet. Sc. Lett., 14, 281-304,

https://doi.org/10.1016/0012-821X(72)90128-8, 1972.





Tropper, P., Manning, C. E., and Essene, E. J.: The substitution of Al and F in titanite at high pressure and temperature: Experimental constraints on phase relations and solid solution properties, J. Petrol., 43, 1787-1814, https://doi.org/10.1093/petrology/43.10.1787, 2002.

Wang, H. Y. C., Chen, H.-X., Lu, J.-S., Wang, G.-D., Peng, T., Zhang, H. C. G., Yan, Q.-R., Hou, Q.-L., Zhang, Q., and Wu, C.-M.: Metamorphic evolution and SIMS U-Pb geochronology of the Qingshigou area, Dunhuang block, NW China: Tectonic implications of the southernmost Central Asian orogenic belt, Lithosphere-US, 8, 463-479, https://doi.org/10.1130/l528.1, 2016.

Wang, H. Y. C., Chen, H.-X., Zhang, Q. W. L., Shi, M.-Y., Yan, Q.-R., Hou, Q.-L., Zhang, Q., Kusky, T., and Wu, C.-M.: Tectonic mélange records the Silurian–Devonian subduction-metamorphic process of the southern Dunhuang terrane, southernmost Central Asian Orogenic Belt, Geology, 45, 427-430, https://doi.org/10.1130/g38834.1, 2017a.

Wang, H. Y. C., Wang, J., Wang, G.-D., Lu, J.-S., Chen, H.-X., Peng, T., Zhang, H. C. G., Zhang, Q. W. L., Xiao, W.-J., Hou, Q.-L., Yan, Q.-R., Zhang, Q., and Wu, C.-M.: Metamorphic evolution and geochronology of the Dunhuang orogenic belt in the Hongliuxia area, northwestern China, J. Asian. Earth. Sci., 135, 51-69, https://doi.org/10.1016/j.jseaes.2016.12.014, 2017b.

Wang, H. Y. C., Zhang, Q. W. L., Chen, H.-X., Liu, J.-H., Zhang, H. C. G., Pham, V. T., Peng, T., and Wu, C. M.: Paleozoic subduction of the southern Dunhuang Orogenic Belt, northwest China: metamorphism and geochronology of the Shuixiakou area, Geodin. Acta., 30, 63-83, https://doi.org/10.1080/09853111.2018.1427407, 2018a.



Wang, H. Y. C., Zhang, Q. W. L., Lu, J.-S., Chen, H.-X., Liu, J.-H., Zhang, H. C. G., Pham,
V. T., Peng, T., and Wu, C.-M.: Metamorphic evolution and geochronology of the
tectonic mélange of the Dongbatu and Mogutai blocks, middle Dunhuang orogenic
belt, northwestern China, Geosphere, 14, 883-906, https://doi.org/10.1130/ges01514.1,
2018b.
Whitney, D.L., and Evans, B.W.: Abbreviations for names of rock-forming minerals, Am.
Mineral., 95, 185-187, https://doi.org/10.2138/am.2010.3371, 2010.
Xu, S. T., Okay, A. I., Ji, S. Y., Sengör, A. M. C., Su, W., Liu, Y. C., and Jiang, L. L.:
Diamond from the Dabie Shan metamorphic rocks and its implication for tectonic
setting, Science, 256, 80-82, https://doi.org/10.1126/science.256.5053.80, 1992.
Ye, K., and Ye, D.: Significance of phosphorous (P)- and magnesium (Mg)-bearing high-
Al titanite in high-pressure marble from Yangguantun, Rongcheng County, Shandong
Province, Chinese. Sci. Bull., 41, 1194-1197,
https://doi.org/CNKI:SUN:JXTW.0.1996-14-009, 1996.
Ye, K., Cong, B., and Ye, D.: The possible subduction of continental material to depths
greater than 200km, Nature, 407, 734-736, https://doi.org/10.1038/35037566, 2000.
Zhang, J., Gong, J., and Yu, S.: c. 1.85 Ga HP granulite-facies metamorphism in the
Dunhuang block of the Tarim Craton, NW China: evidence from U–Pb zircon dating
of mafic granulites, J. Geol. Soc. London., 169, 511-514,
https://doi.org/10.1144/0016-76492011-158, 2012.
Zhang, J., Yu, S., and Mattinson, C. G.: Early Paleozoic polyphase metamorphism in
northern Tibet, China, Gondwana. Res., 41, 267-289,
https://doi.org/10.1016/j.gr.2015.11.009, 2017.



Zhang, Q. W. L., Wang, H. Y. C., Liu, J.-H., Shi, M.-Y., Chen, Y.-C., Li, Z. M. G., and
Wu, C.-M.: Diverse subduction and exhumation of tectono-metamorphic slices in the
Kalatashitage area, western Paleozoic Dunhuang Orogenic Belt, northwestern China,
Lithos, 360-361, https://doi.org/10.1016/j.lithos.2020.105434, 2020.
Zhao, Y., Sun, Y., Diwu, C., Guo, A.-L., Ao, W.-H., and Zhu, T.: The Dunhuang block is
a Paleozoic orogenic belt and part of the Central Asian Orogenic Belt (CAOB), NW
China, Gondwana. Res., 30, 207-223, https://doi.org/10.1016/j.gr.2015.08.012, 2016.
Zong, K. Q., Zhang, Z. M., He, Z. Y., Hu, Z. C., Santosh, M., Liu, Y. S., and Wang, W.:
Early Palaeozoic high-pressure granulites from the Dunhuang block, northeastern
Tarim Craton: constraints on continental collision in the southern Central Asian
Orogenic Belt, J. Metamorph. Geol., 30, 753-768, https://doi.org/10.1111/j.1525-
1314.2012.00997.x, 2012.














**TABLE 1**. PRESSURE-TEMPERATURE (P-T) CONDITIONS RETRIEVED FOR
THE DIFFERENT METAMORPHIC STAGES OF GARNET CLINOPYROXENITE

| Sample | Prograde assemblage (M1) | | | Peak assemblage (M2) | | | Retrograde assemblage (M3) | | |
|--------|---------|----------|--------|---------|----------|--------|---------|----------|--------|
| | $T$ (°C) | $P$ (kbar) | Method | $T$ (°C) | $P$ (kbar) | Method | $T$ (°C) | $P$ (kbar) | Method |
| 17D78 | 662 | 5.4 | Hbl | 789 | 28 | $GC_{12}$ | 621 | 4.6 | Hbl |
| 17D80 | - | - | | 902 | 38.2 | $GC_{12}$ | 656 | 5.4 | Hbl |
| 17D90 | 695 | 7.2 | Hbl | 868 | 31.8 | $GC_{12}$ | 669 | 6 | Hbl |
| 17D95 | - | - | | 918 | 41.3 | $GC_{12}$ | 631 | 5.6 | Hbl |

Note: Geothermobarometry symbols are given in footnotes.
Hbl is the monomineralogic hornblende geothermobarometers (Gerya et al., 1997).
GC12 is the garnet-clinopyroxene geothermometer (Nakamura, 2009) coupled with the
garnet-clinopyrxene geobarometer (Beyer et al., 2015).


















**Figure Captions**


**Figure 1.** (A) Sketch map showing the Central Asian orogenic belt and adjacent
cratons (modified after Han et al., 2015). (B) Tectonic sketch of the Dunhuang orogenic
belt and its surrounding tectonic units (modified after Zhang et al., 2017).
**Figure 2.** Tectonic sketch of the Dunhuang orogenic belt (modified after Lu et al.,
2006; Zhang et al., 2017).
**Figure 3.** (A) Geologic map of the Mt. Sanweishan area (modified after 1:1,000,000
geological map of Gansu Province). (B) Geological map of the granitic pluton.
**Figure 4.** Outcrops of the garnet clinopyroxenite.
**Figure 5.** Micropetrography of the garnet clinopyroxenite. Subscripts 1, 2, 3 and 4
refer to the prograde (M1), metamorphic peak (M2), first retrograde (M3) and final
retrograde (M4) minerals, respectively. The dashed red arrow refers to electron microprobe
analytical profile of garnet. Mineral abbreviations are after Whitney and Evans (2010). (A)
The prograde assemblage (M1) is the tiny inclusions $Hbl_1$ + $Ilm_1$ preserved in the garnet
interior. The peak metamorphic assemblage (M2) consists of $Grt_2$ + $Cpx_2$ + $Ilm_2$. The first
retrograde assemblage (M3) is the symplectic $Hbl_3$ + $Pl_3$ intergrowth formed in between
the matrix $Grt_2$ and $Cpx_2$. (B) Besides the M1, M2 and M3 assemblages similar to that in
(a), the $Cpx_2$ rim partially retrograded to $Hbl_3$. (C) The symplectic assemblage (M3) $Hbl_3$
+ $Pl_3$ formed in between the matrix $Grt_2$ and $Cpx_2$. (D) The retrograde assemblage (M3)
$Hbl_3$ + $Pl_3$ + $Bt_3$ formed both in the $Grt_2$ interior and in between the matrix $Grt_2$ and $Cpx_2$,
and the retrograde $Chl_4$ formed from the $Hbl_3$ rim. (E) The retrograde minerals $Pl_3$, $Ilm_3$
and $Ttn_3$ lamellae (M3) exsolved from within the $Cpx_2$. (F) The $Cpx_2$ rim retrograded to
$Hbl_3$. (G) Most of the $Rt_2$ retrograded to $Ilm_3$. (H) The $Grt_2$ was almost completely



retrograded to Hbl$_3$ + Pl$_3$, and Act$_4$ formed from the Hbl$_3$ rim. (I) The Hbl$_3$ and Ilm$_3$
lamellae (M3) exsolved from within the Cpx$_2$.
**Figure 6.** Micropetrographic evidences of UHP metamorphism. (A) High-Al titanite
enclosed in garnet porphyroblast. (B) High-Al titanite rimmed by ilmenite and hornblende
within garnet porphyroblast. (C-D) At least three groups of rutile lamellae (the needles)
exsolved from within the garnet. (E) Idiomorphic multiple-phase inclusion of spinel (Spl)
+ tremolite (Tr) ±rutile (Rt) within garnet. (F) Idiomorphic hexagon ilmenite separated by
high-Al titanite from midcourt line in garnet.
**Figure 7.** X-ray compositional mapping of MnO, MgO, FeO, and CaO components
of representative garnet porphyroblast in samples 17D80, 17D90, and 17D95.
**Figure 8.** Chemical compositional profiles of the garnet porphyroblast in samples
17D78, 17D80, 17D90, and 17D95.
**Figure 9.** Metamorphic *P-T* paths of the four garnet clinopyroxenite samples. The
boundaries of metamorphic facies and metamorphic facies series are taken from O'Brien
and Rötzler (2003) and Spear (1993), respectively. The diamond = graphite and quartz =
coesite polymorph transition curves are taken from Kennedy and Kennedy (1976) and Bose
and Ganguly (1995), respectively.
**Figure 10.** U-Pb concordia diagram of analyzed titanite separated from the garnet
clinopyroxenite samples 17D78, 17D95 and 17D90. Data are plotted at 2σ level, and
uncertainties on lower intercept ages are on the 95% confidence.



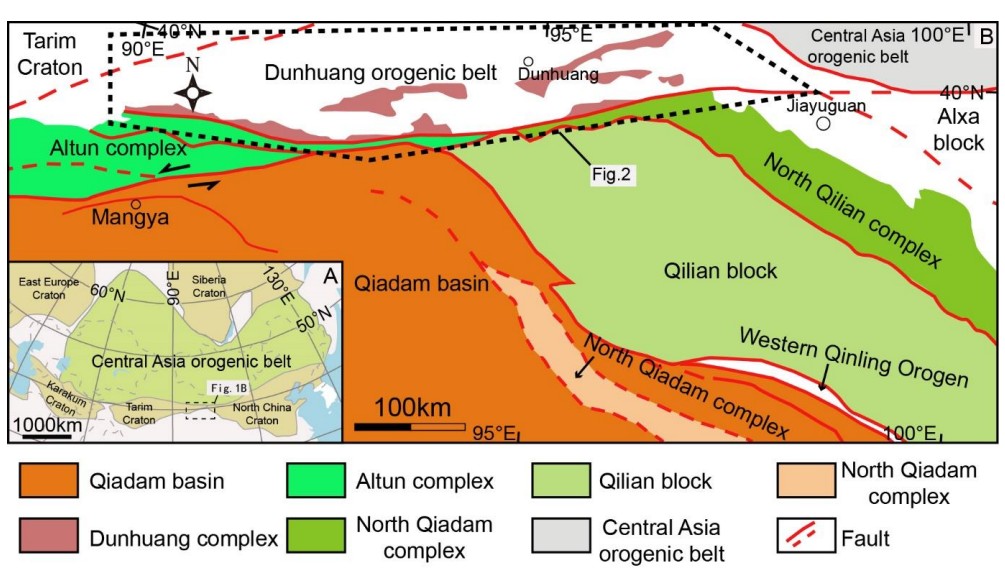


Figure 1


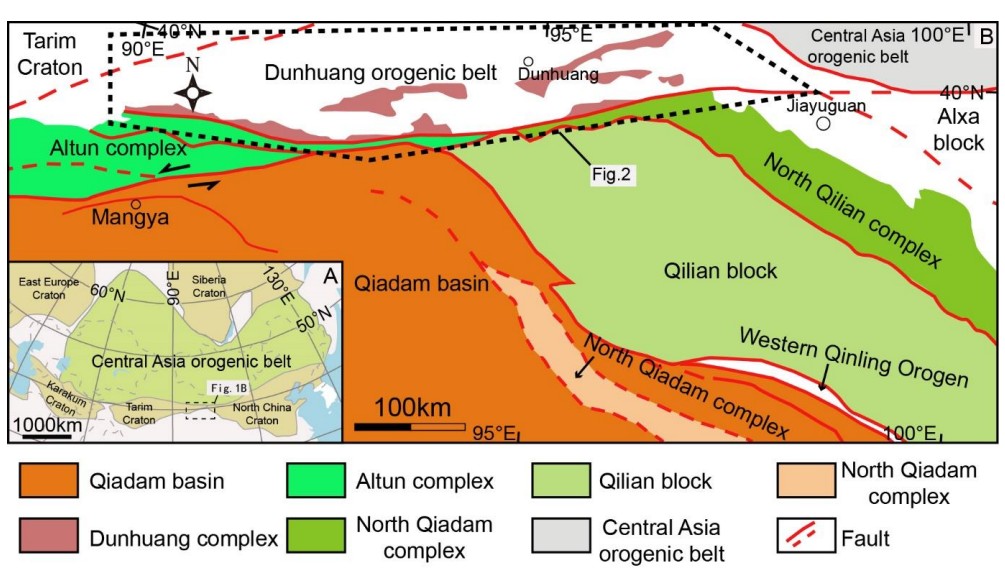




717          Figure 2







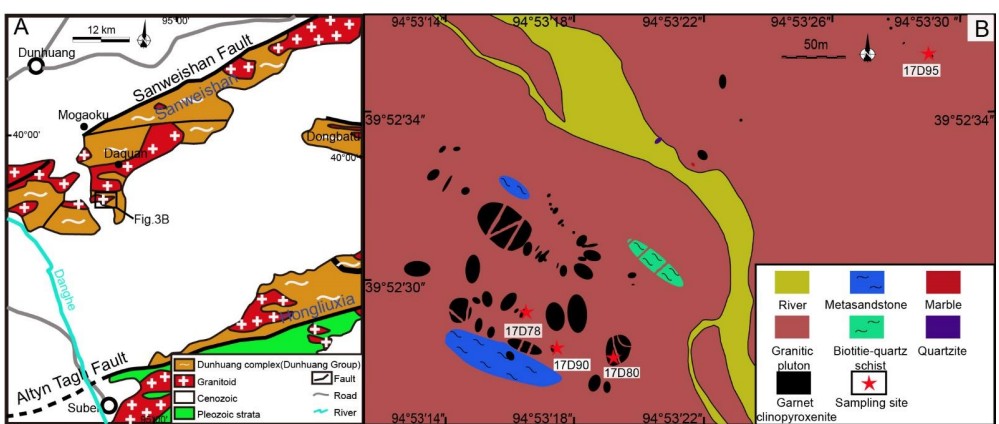




Figure 3


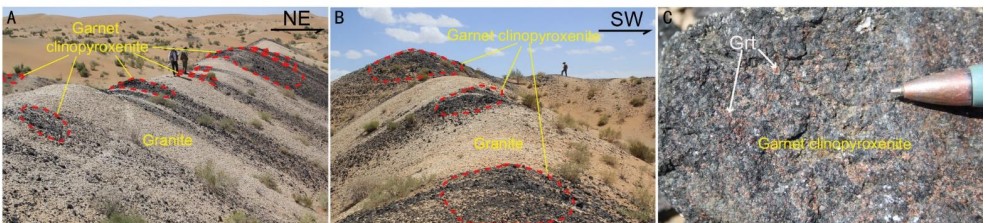




Figure 4




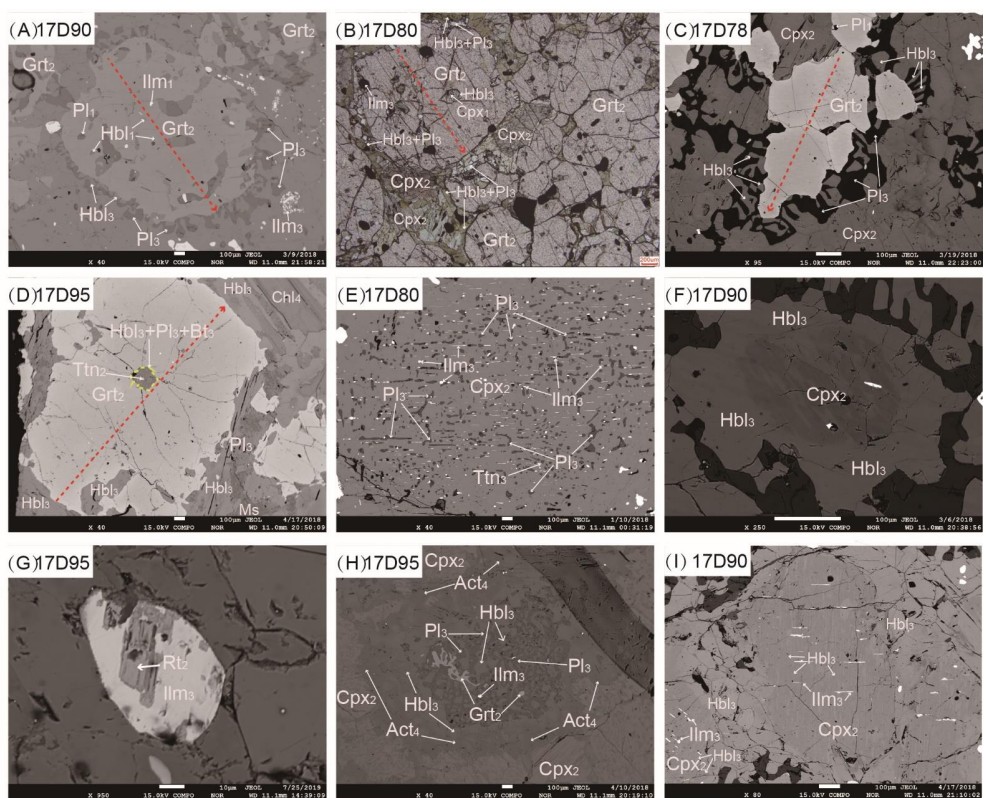



730                                    Figure 5








734                          Figure 6








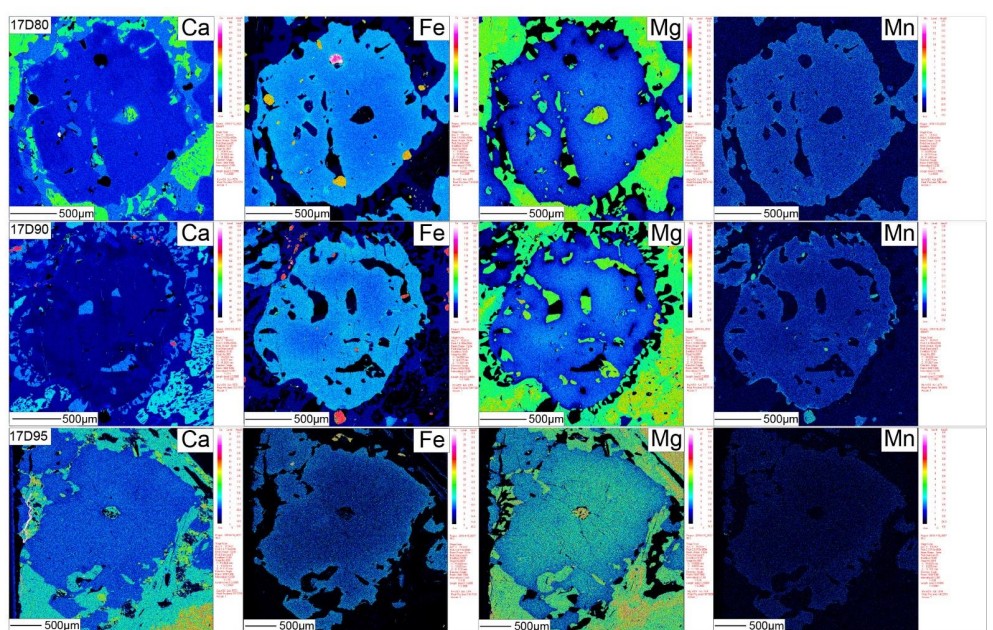



740                                         Figure 7
















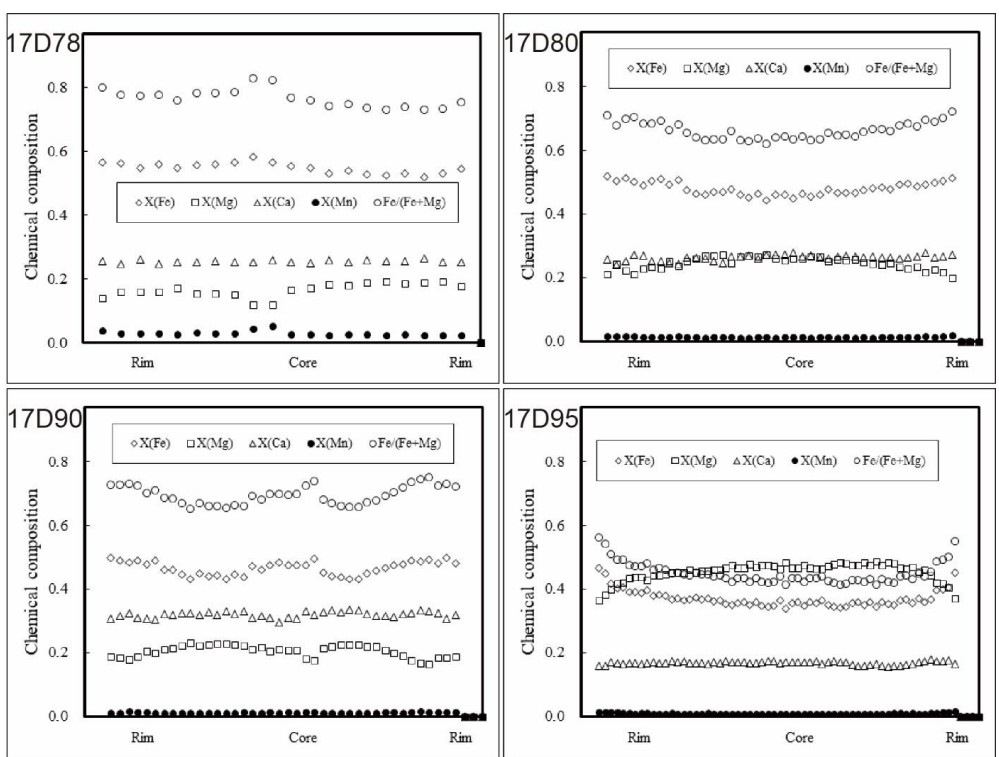

Figure 8




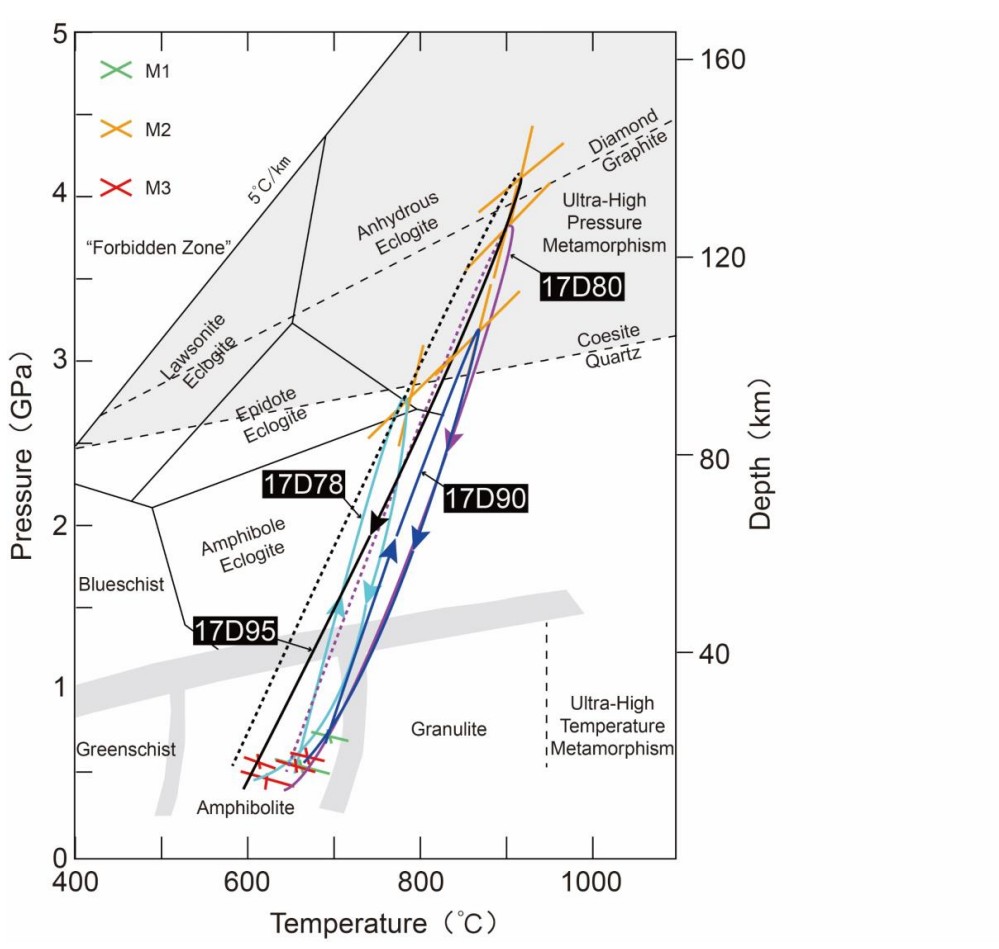



766                                    Figure 9









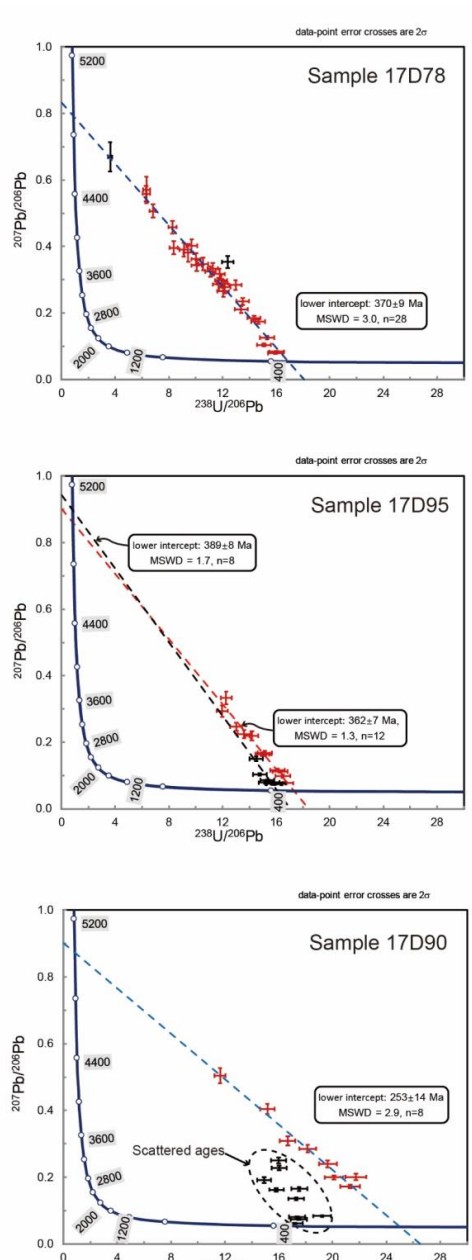



Figure 10