# Peer review of "First report of ultra-high pressure metamorphism in the Paleozoic Dunhuang orogenic belt (NW China): Constrains from *P-T* paths of garnet clinopyroxenite and SIMS U-Pb dating of titanite"

_Solid Earth, 2020_

## Referee Comment (RC1) · Anonymous Referee #1 · 28 Jul 2020

**Review of 'First report of ultra-high pressure metamorphism in the Paleozoic Dunhuang orogenic belt (NW China): Constraints from P-T paths of garnet clinopyroxenite and SIMS U[Pb dating of titanite"** //

**Summary** This manuscript presents results from garnet clinopyroxenite enclaves in the Dnhuang Orogenic Belt, southwest China. The authors produce a comprehensive and thorough microstructural characterization of successive mineral assemblages, upon which geochemical analyses are built to construct a quantitative thermobarometric history of the P–T conditions prograde, peak, and retrograde metamorphism. Application

of conventional thermobarometry. crystallographically controlled exsolution of rutile, and the high Al content of titanite is used to suggest that these enclaves were subducted as part of a tectonic melange to high pressures of 28–41 kbar, well within the UHP metamorphic regime. SIMS U-Pb titanite dates of c. 390–370 Ma are interpreted as post-peak cooling ages associated with exhumation. Below, I present several general comments and my recommendations for the manuscript, in addition to some more detailed line-by-line comments encompassing those suggestions.

**General comment**

Overall, this is an interesting paper that presents nice microstructural observations, a wealth of geochemical data, and some well-constrained ages. I enjoyed reading it. The manuscript is succinct, if a little lacking in detail and discussion in places. Below, I have pointed out places that I feel rephrasing is necessary to ensure clarity, and a few technical corrections, but an additional careful language edit would be valuable to ensure total clarity of the scientific content and ideas. The presentation of a new UHP setting is very exciting and certainly appropriate for publication in Solid Earth, and the authors do a good job of using the relatively scarce literature pertaining to UHP metamorphism preserved in lithologies other than eclogites. However, as the submission proposes to identify a new UHP locality—certainly a significant and important contribution to a wide range of geoscientists—I have a couple main issues that I believe need to first be addressed that would ensure that the authors' interpretation is robust and watertight.

Much of the interpretation of UHP conditions of peak metamorphism comes from the presence of exsolved rutile needles in clinopyroxene. As the authors discuss, this may indeed be a consequence of decompression from UHP, but it is not ubiquitous or uniquely diagnostic of this setting. The same process can happen during cooling from high-T granulite facies metamorphism (e.g., in garnet clinopyroxenites in the Jijal Complex of the Kohistan Arc). Given that the compositions of clinopyroxene (neither the analyzed nor the recalculated) is jadeitic, the possibility of this having formed during cooling from the granulite facies to the amphibolite facies should be discussed in more

detail. The authors did not much discuss the host granitic body, or the nature of the contacts between this host and the enclaves, which might also have a bearing on the genesis of such clinopyroxenite bodies (HP and UHP enclaves interpreted as mélange units are often hosted in softer metasediments, rather than granite).

As clinopyroxene are not jadeitic and the compositions used for conventional thermobarometery are recalculated (the methodology of which should also be detailed), I recommend the use of additional thermobarometers to provide additional confirmation of UHP metamorphism. Trace element thermobarometers (several of which are detailed below), may be a little more reliable in that trace element diffusivities are slower than Fe-Mg at high-T, and are thus less sensitive to diffusional homogenization and modification.

**Specific comments**

L53: Consider rephrasing "in the subduction process" to "during subduction"

L57: Other potential indicators could include acicular inclusions of rutile and pyroxene in metapelitic garnet (Ague and Eckert, 2012), quartz rods and needles in omphacite indicate an exsolution from a supersilicic clinopyroxene that contained a Ca‐Eskola component (JAnák et al, 2004), or the phengite content of white mica?

L58–59: Please consider rephrasing this sentence.

L61: Is the host lacking in any kind of metamorphic fabric? Woah! If this is the case, how do you explain this?

L63–64: References for the evolution of our understanding of the Dunhuang area would be valuable here.

L83: What evidence is there (or who suggests) that dissection of the block by faults was "possibly in the Tertiary"?

L88: "features"

L90: It's not completely obvious to someone unfamiliar with the setting why the slices were metamorphosed at different depths. How different are these depths? Some references here would be valuable.

L92: Suggest removing "unfortunately," as it implies that your pre-determined conclusion was to identify UHP metamorphism. Instead, better to frame your discussion in terms of the global scarcity of UHP indicator minerals, and whether this reflects the geologically rare set of circumstances required for UHP metamorphism, or if their scarcity reflects a preservation bias.

L92: Recommend replacing "inter-granular" with "matrix" and "inclusions"

L95: As in Franz & Spear (1985), I recommend you reference the alternative nomenclature: sphene

Petrography Section: Perhaps you could describe the field relationships a little more extensively? I would be interested in knowing the nature of the contacts between the enclaves and the granite (conformable, sharp etc). From your Figure 4, it looks as if the enclaves are broadly coherent, meters wide, and aligned parallel to some internal granitic foliation?

L112: In text you refer to Ttn1, but in Figure 6a and 6b it's potentially labelled Ttn2 (I can't find a Ttn1 in these figures). Is the Titanite part of the peak or prograde assemblage?

L114–115: Can these accessory phases be seen in Figure 5 at all?

L116: "worm-like"

L118: "rimming"

L119–121: Yes, this texture has been documented to reflect exhumation from high-pressure conditions to the amphibolite facies if the clinopyroxene is jadeitic. However, if the clinopyroxene is more augitic in composition it may reflect cooling from the granulite

to amphibolite facies.

L120–128: A really interesting set of observations!

L133: The Hwang et al. (2019) paper only discusses rutile exsolution in garnet; is there precedent for spinel, tremolite and rutile exsolving from garnet? I'm not sure – might be worth doing a mass balance to find out if it's geochemically possible in a closed system! Alternatively, could these be some kind of late recrystallized inclusion or something associated with fluid infiltration along a crack?

L135: Consider rephrasing "midcourt." In Figure 6f, this inclusion looks like it's connected to a fracture. Could it be fluid related?

L147: Please consider replacing "tiny" with smallest

L149: Do you have a record of where the analyses were taken, e.g., cores vs rims of grains?

L155: Please define how you calculate garnet end-members. Is this variation quoted between samples? Is within sample variation in garnet composition negligible?

L156–157: What chemistry would you expect in mantle xenoliths or eclogite? Please be a little more explicit, as it's not necessarily common knowledge amongst all Solid Earth readers.

L159–161: There looks to be a slight Mn peak in the rim of 17D90 and 17D80 also consistent with a little bit of resorption. Be explicit in terms of the petrographic evidence for resorption.

S1 and S2: Please include recalculated cations in data tables. S1 would be useful as a standard table in the main text, rather than as supplementary information.

L164–165: "the chemical composition" Also here, please clarify what you mean by "altered to a different extent." Different to what?

L167–168 and L175: Your calculations are highly dependent on these 'reintegrated' compositions, so please provide some information about the approach taken to calculate these? Did you estimate the proportion of exsolution lamellae and reincorporate their chemistry according to this mode?

L168–170: Additional references here could include Carswell et al, Min. Mag (1996) and Castelli, JMG (1991). It should also be noted that these references and those in your current manuscript only document high-Al and F titanite in marbles and calcsilicates, if your measured Al2O3 reflects high P metamorphism (your concentrations are significantly less than those in Franz & Spear, 1985, I believe), your manuscript would (though I'm admittedly not that familiar with the literature so there may be those I'm not aware of) be the first documentation of such compositions in garnet pyroxenites. Frost et al., Chemical Geology (2001) have a good discussion of the expected metamorphic stability of sphere/titanite in metabasites, and because your results extend their general observations, I recommend expanding your discussion to include this paper.

Section: Geothermobarometry: You make some good observations, but it would be useful if there was more evidence for the peak conditions you appeal to, as I think they could be interpreted as having formed during decompression from high-T. I recommend you include the Zr-in-sphene/titanite (Hayden et al, CMP, 2008) thermobarometer which has been shown to sensitive to pressure variations and is calibrated up to 1000 ËŽC with ±20 ËŽC, If coupled with the Zr-in-rutile thermometer and potentially the Ti-in-rutile thermometer (e.g., see Ewing et al., CPM, 2012 and Liu et al., Gondwana Research, 2015) you could provide additional independent constraints on peak P-T. You also have quite a lot of geochemical data, did you consider applying pseudosection thermobarometry (e.g, in Theriak-Domino, Thermocalc or Perplex) or the AveragePT approach in Thermocalc (especially now that there are recently updated models for your chemical system)? For the former, you might get quite large, low variance fields, but it might be a useful exercise to see compare with your more

conventional barometers.

L174: I don't think it's appropriate to simply use the average chemical composition of garnet with matrix clinopyroxene for conventional thermobarometry to obtain conditions of peak metamorphism unless garnet chemistries are completely homogenous, and you think this homogenization occurred at peak. If you make the assumption that this is the case, it is worth being explicit with this. Your garnets in 17D80 and 17D95 are approximately homogeneous, but rim chemistries might have been incorporated during cooling. You should not use these rim chemistries at all, and only use plateau cores. There is some Mn zoning preserved in the core of 17D78! I would suggest being a little more targeted with the use of these chemistries when using them with conventional thermobarometers.

L179: I recommend you include the error estimates ($\pm 4$ kbar) in your Figure 9.

L179–180: "...assemblages mainly consist of plagioclase..."

L182: Please include errors with hornblende thermobarometers!

L192: I don't think you've necessarily "certified UHP metamorphism." Generally, this is only really truly the case if coesite pseudomorphs or marjoritic garnet is documented. While your data might well suggest this as it currently stands, your results are highly dependent on results of a single barometer which uses chemistries that are highly modified to their present state. I also find it strange that your jadeite component in clinopyroxene is so low in the chemistry you infer as having equilibrated at UHP conditions; why might this be?

L193–194: Exsolution of rutile lamellae in garnet is not exclusive to decompression from UHP metamorphism (as far as I am aware). It can also be documented as having formed during the retrogression of mafic garnet granulites from high-T conditions so is not necessarily diagnostic.

L203: Remove "scenario"

Dating metamorphism and L226: Please include geochronological data tables in the main text, rather than as supplementary tables! Are the ages you quote weighted averages? Also, in Supplementary Table 4 you quote uncertainties as "±s" which is a good convention to follow; are the ± intervals quoted in the text also s, rather than sigma?

L221: Please correct to "Terra-Wasserburg" Nice ages!!

L221–224: That's ok, but if titanite formed at the expense of a high-$\mu$ phase (e.g., rutile) then it will most likely incorporate more radiogenic Pb than the bulk rock, potentially causing problems with your common Pb projection. Any thoughts on what reactions may have formed titanite?

L227Âř–232: The titanites in 17D90 and 17D95 are altered? Do you mean by diffusion, or something else like fluid fluxing? The zoning in the BSE looks to me to be primary, perhaps? In places, it's concentric, and the sharp boundaries are consistent with negligible diffusional homogenization. Did you consider trying to put more multiple spots down on individual grains? In a grain in 17D95 you've done this, and the core age gives a nearly 20 Myr younger age than the rim, which is interesting/strange and potentially worth investigating! Patchy zoning such as this has been related to interface coupled dissolution and precipitation reactions (Bonamici et al, 2015; Garber et al, 2015; Walters Kohn, 2017), so your observations might provide a nice framework. See next comment for more on this.

L234–236: While you're correct that titanite U-Pb ages are traditionally thought to record cooling through 600Âř–750 ËŽC (additional references to those you've provided include Cherniak (1993) and Spear and Parrish, (1996)), recent studies (including experimental studies) have indicated that Pb diffusion seems to be negligible in titanite at T < 850 ËŽC. Some relevant examples include Castelli and Rubatto, CMP, 2002; Garber et al, J Pet, 2017; Walters and Kohn, JMG, 2017; Holder and Hacker, Chemical Geology, 2019. Rather than cooling, these authors document U–Pb titanite

dates that record the timing of neo-crystallization, fluid-driven alteration/resetting, or deformation that might facilitate resetting of U-Pb dates and compositions. It is worth considering potential alternative scenarios given increasing evidence that dates such as those you've obtained in this study may not reflect cooling, but some other process.

L242–243: Maybe replace the end of this sentence with "assess the likelihood of UHP metamorphism"?

L255–258: I'm still not entirely sure why "the prograde assemblage (M1) clearly indicates the subduction process". Please expand on this. "Very deep but different depths" is not clear, either. Please clarify.

L245–250: This paragraph should be placed in a little more context to account for its inclusion.

L274–278: I recommend you rephrase this paragraph, but yes, I think the sample could alternatively reflect decompression from high-T conditions (as mentioned in a previous comment), based at least on the microstructures. If there are other barometers that you could apply to this dataset to evidence your UHP conclusion, great! I think the possibility that your pyroxene compositions are not reflective of UHP conditions and instead reflect HT metamorphism should be explored a little more. Would this interpretation potentially fit with the garnet clinopyroxenite field setting (in the undeformed granitic bodies) a little better, too?

L280–282: What about the potentially important role of intracrystalline diffusional exchange (e.g., Pattison Begin, 19094) that results in Fe-Mg exchange at the relatively high temperatures you suggest? Geothermobarometers that use net transfer reactions and Ca, Al and Si tend to be a little more reliable at these conditions.

L293: By "wide representative," do you mean its applicable to a wide range of bulk rock compositions?

L315: Here and in a previous paragraph, you justify use of the geothermobaromers

as your chosen calibrations are "most accurate." I would recommend modifying this discussion, because results are only accurate if you know the true result, how do you know the calibration is accurate for your bulk composition? Has it been calibrated for such (in which case, great!!).

L335–338: A subduction mélange? Also recommend changing "…large gaps of the peak…" to "…large differences in the peak…"

L345: consider rephrasing "which is neglected to different extent in the past"

———————————————

---

## Referee Comment (RC2) · Chunjing Wei (Referee) · 31 Jul 2020

Ultra-high pressure (UHP) metamorphic rocks of continental affinity indicate that continental slabs can subduct to great depths where coesite and diamond can stabilize, and UHP metamorphism has been a hot topic for the past decades. Li et al. (2020) for the first time report UHP garnet clinopyroxenites from the Paleozoic Dunhuang orogenic belt, NW China. The rocks are retrieved to show clockwise P–T paths with the peak conditions of 790~920 °C / 28~41 kbar that are constrained by available garnet-clinopyroxene thermobarometries. This UHP metamorphism can be further

confirmed by the presence of high-Al titanite inclusions in garnet and pyrope-rich garnet with exsolved rutile lamellae. Titanite SIMS U-Pb dating yields a metamorphic age of 389~370 Ma, interpreted to represent the post peak exhumation time. The evidence for the UHP metamorphism is robust and the age data are in good quality. It will be much better to involve available bulk-rock compositions for both major and trace elements because these are significant for understand the petrogenetic origin of the UHP rocks. The discovery of the UHP garnet clinopyroxenites is great advance for the study of the Dunhuang orogenic belt, which has added a new case for the globe occurrences of UHP metamorphic rocks.

Comments from Chunjing Wei at the School of Earth and Space Sciences, Peking University, China. cjwei@pku.edu.cn

―――――――――――

---

## Author Comment (AC1) · 7 Sep 2020

**Review of 'First report of ultra-high pressure metamorphism in the Paleozoic Dunhuang orogenic belt (NW China): Constraints from P-T paths of garnet clinopyroxenite and SIMS U-Pb dating of titanite" //**

**Summary**

This manuscript presents results from garnet clinopyroxenite enclaves in the Dnhuang Orogenic Belt, southwest China. The authors produce a comprehensive and thorough microstructural characterization of successive mineral assemblages, upon which geochemical analyses are built to construct a quantitative thermobarometric history of the P–T conditions prograde, peak, and retrograde metamorphism. Application of conventional thermobarometry. crystallographically controlled exsolution of rutile, and the high Al content of titanite is used to suggest that these enclaves were subducted as part of a tectonic melange to high pressures of 28–41 kbar, well within the UHP metamorphic regime. SIMS U-Pb titanite dates of c. 390–370 Ma are interpreted as post-peak cooling ages associated with exhumation. Below, I present several general

comments and my recommendations for the manuscript, in addition to some more detailed line-by-line comments encompassing those suggestions.

**General comment**

Overall, this is an interesting paper that presents nice microstructural observations, a wealth of geochemical data, and some well-constrained ages. I enjoyed reading it. The manuscript is succinct, if a little lacking in detail and discussion in places. Below, I have pointed out places that I feel rephrasing is necessary to ensure clarity, and a few technical corrections, but an additional careful language edit would be valuable to ensure total clarity of the scientific content and ideas. The presentation of a new UHP setting is very exciting and certainly appropriate for publication in Solid Earth, and the authors do a good job of using the relatively scarce literature pertaining to UHP metamorphism preserved in lithologies other than eclogites. However, as the submission proposes to identify a new UHP locality—certainly a significant and important contribution to a wide range of geoscientists—I have a couple main issues that I believe need to first be addressed that would ensure that the authors' interpretation is robust and watertight.

Much of the interpretation of UHP conditions of peak metamorphism comes from the presence of exsolved rutile needles in clinopyroxene. As the authors discuss, this may indeed be a consequence of decompression from UHP, but it is not ubiquitous or uniquely diagnostic of this setting. The same process can happen during cooling from high-T granulite facies metamorphism (e.g., in garnet clinopyroxenites in the Jijal Complex of the Kohistan Arc). Given that the compositions of clinopyroxene (neither the analyzed nor the recalculated) is jadeitic, the possibility of this having formed during cooling from the granulite facies to the amphibolite facies should be discussed in more detail. The authors did not much discuss the host granitic body, or the nature of the contacts between this host and the enclaves, which might also have a bearing on the genesis of such clinopyroxenite bodies (HP and UHP enclaves interpreted as mélange

units are often hosted in softer metasediments, rather than granite).

[REPLY] Thanks for your suggestions. The Cpx is not jadeitic. The garnet clinopyroxenite is essentially enclave enclosed in the granitic body, which contact is clear and abrupt. As for the UHP diagnostics and the possibility of cooling from the granulite facies to the amphibolite facies, are answered in "Specific comments and Reply" of Lines 119-121, Line 192 and Lines 193-194 in detail. The question about the relation between garnet clinopyroxenite and its host granitic body is answered in detail in "Specific comments and Reply" of Line 61 and "Petrography Section".

As clinopyroxene are not jadeitic and the compositions used for conventional thermo-barometery are recalculated (the methodology of which should also be detailed), I recommend the use of additional thermobarometers to provide additional confirmation of UHP metamorphism. Trace element thermobarometers (several of which are detailed below), may be a little more reliable in that trace element diffusivities are slower than Fe-Mg at high-T, and are thus less sensitive to diffusional homogenization and modification.

[REPLY] Thank you. In fact, distributions of trace elements among minerals can be calibrated as geothermometers but NOT geobarometers, because pressure effects on trace elemental distributions is negligible. We note the methodology of recalculation of clinopyroxene composition in Lines 848-850 and Lines 859-861 now. About the trace element thermobarometers, we answered in detail in "Specific comments and Reply" of "Section: Geothermobarometry".

**Specific comments**

L53: Consider rephrasing "in the subduction process" to "during subduction"

[REPLY] Thank you. We have changed this phrase in the new manuscript (now in Line

53).

L57: Other potential indicators could include acicular inclusions of rutile and pyroxene in metapelitic garnet (Ague and Eckert, 2012), quartz rods and needles in omphacite indicate an exsolution from a supersilicic clinopyroxene that contained a Caˇ A ˇ REskola component (JAnák et al, 2004), or the phengite content of white mica?

[REPLY] Thank you. Yeah, we cannot agree with you more. There are so many indicators of UHP metamorphism (except you list above, e.g., K-rich clinopyroxene, majoritic garnet, monozite lamellae exsolved from apatite, supersilicic titanite, and others see review in You et al., 2007, Geoscience (in Chinese with English abstract)), so we only list the most common UHP indicators in this manuscript (now in Lines 54-58).

L58–59: Please consider rephrasing this sentence.

[REPLY] Thank you. We have rephrased this sentence (now in Lines 58-60).

L61: Is the host lacking in any kind of metamorphic fabric? Woah! If this is the case, how do you explain this?

[REPLY] Thank you. In fact, we can't see any foliation or other metamorphic fabric of the granitic pluton both in the field and under microscope. In fact, this pluton could not be involved in the subduction process (thus it is undeformed and unmetamorphosed); actually, it might intrude the crust and afterwards it brought the enclaves (including garnet clinopyroxenite and paragneiss) up from the deeper to shallower crustal level. Therefore, the UHP rocks were also hosted in softer metasediments before they were captured by their present host (the granitic pluton) (now in Lines 100-104). We have tried to date this pluton, but unfortunately, we failed due to severe decrystallization of magmatic zircon, possibly caused by radioactive damage (now in Lines 104-105).

L63–64: References for the evolution of our understanding of the Dunhuang area would be valuable here.

[REPLY] Thank you. Some related references have been listed in the manuscript (now in Lines 64-65).

L83: What evidence is there (or who suggests) that dissection of the block by faults was "possibly in the Tertiary"?

[REPLY] Thank you. This interpretation is based on our field observations and remote sensing analysis of Cunningham et al (2016) (now in Lines 84-85).

L88: "features"

[REPLY] Thank you. We have changed this word to plural form (now in Line 90).

L90: It's not completely obvious to someone unfamiliar with the setting why the slices were metamorphosed at different depths. How different are these depths? Some references here would be valuable.

[REPLY] Thank you. We consider the slices were metamorphosed at different depths because of the differences in peak pressure (which could be comparable to depth, e.g., Wang et al., 2017a; Zhang et al., 2020; this manuscript) (now in Lines 91-95).

L92: Suggest removing "unfortunately," as it implies that your pre-determined conclusion was to identify UHP metamorphism. Instead, better to frame your discussion in terms of the global scarcity of UHP indicator minerals, and whether this reflects the geologically rare set of circumstances required for UHP metamorphism, or if their scarcity reflects a preservation bias.

[REPLY] Thank you. We have rephrased this sentence and explained the reasons of difficulty in recognition of UHP metamorphism (now in Lines 96-98).

L92: Recommend replacing "inter-granular" with "matrix" and "inclusions"

[REPLY] Thank you. We have rephrased this sentence (now in Lines 96-98).

L95: As in Franz & Spear (1985), I recommend you reference the alternative nomenclature: sphene

[REPLY] Thank you. We consider there is no substantial differences between sphene and titanite, and we found titanite seems more prevail in recent years, so we chose the latter (now in Line 98).

Petrography Section: Perhaps you could describe the field relationships a little more extensively? I would be interested in knowing the nature of the contacts between the enclaves and the granite (conformable, sharp etc). From your Figure 4, it looks as if the enclaves are broadly coherent, meters wide, and aligned parallel to some internal granitic foliation?

[REPLY] Thank you. The relationship between the enclaves and the granite is structural contact (sharp, as demonstrated now in Lines 100-104). We have to clarify that the enclaves are not aligned parallel to nonexistent internal granitic foliation or broadly coherent; the real locality of garnet clinopyroxenite is limited in the red dotted ring (Figs. 4A-4B).

L112: In text you refer to Ttn1, but in Figure 6a and 6b it's potentially labelled Ttn2 (I can't find a Ttn1 in these figures). Is the Titanite part of the peak or prograde assemblage?

[REPLY] Thank you. Sorry, it's a clerical error. In fact, we consider all the Al-rich titanite formed in the peak metamorphism, so it should be $Ttn_2$ (now in Line 118).

L114–115: Can these accessory phases be seen in Figure 5 at all?

[REPLY] Thank you. We feel very sorry for that no apatite or magnetite could be seen in Figure 5, but we recognized them under electron microprobe, so we have changed our expression (now in Lines 119-121).

L116: "worm-like"

[REPLY] Thank you. We have rephrased this word (now in Line 123).

L118: "rimming"

[REPLY] Thank you. We have rephrased this word (now in Line 124).

L119–121: Yes, this texture has been documented to reflect exhumation from high-pressure conditions to the amphibolite facies if the clinopyroxene is jadeitic. However, if the clinopyroxene is more augitic in composition it may reflect cooling from the granulite to amphibolite facies.

[REPLY] Thank you. Clinopyroxene in each sample is essentially diopside based on the classification of Morimoto (1988) (now in Lines 271-273), and there are many examples of decompression where the diopside preserved. What's more, if amphibolite facies assemblage retrograded from high-pressure granulite, it should also represent decompression (e.g., Wang et al., 2018b; Zhang et al., 2020). Instead, "red-eye socket" symplectites might be formed while cooling from medium-pressure granulite to amphibolite facies and where orthopyroxene would be preserved too (e.g., Zhang et al.,

2019, Precambrian. Res). In this work, decompression has been demonstrated by geothermobarometry.

L120–128: A really interesting set of observations!

[REPLY] Thank you. They are now in Lines 128-137.

L133: The Hwang et al. (2019) paper only discusses rutile exsolution in garnet; is there precedent for spinel, tremolite and rutile exsolving from garnet? I'm not sure–might be worth doing a mass balance to find out if it's geochemically possible in a closed system! Alternatively, could these be some kind of late recrystallized inclusion or something associated with fluid infiltration along a crack?

[REPLY] Thank you. In fact, in our knowledge, there was no precedent for spinel, tremolite and rutile assemblage together exsolved from garnet. However, other kinds of minerals were found to be exsolved from garnet (e.g., clinopyroxene, rutile, apatite, Ye et al., 2000), which indicate that garnet could contain many other elements except for Ca, Fe, Mg, Mn and Al while the garnet was formed in extreme condition (e.g., Na, P, Ti, Ye et al., 2000). The most important thing is that the assemblages (spinel, tremolite and rutile) are idiomorphic, while the inclusion minerals formed in the prograde metamorphism are round in shape; if there was a crack, the shape of the assemblage would be constrained by the crack. In fact, prior to make thin sections, cutting of the rock is random in directions and actually no fractures were found. Therefore, we prefer they were exsolved from garnet (now in Lines 137-142).

L135: Consider rephrasing "midcourt." In Figure 6f, this inclusion looks like it's connected to a fracture. Could it be fluid related?

[REPLY] Thank you. We have replaced "midcourt line" by "centerline". Yes, there is a crack crossing the inclusion, but interestingly, we found the same phenomenon in

Ague and Eckert (2012, their figure 3c, see below), so we just made an analogy. Of course it might be fluid related, but we don't have evidence, either. Due to it's idiomorphic hexagon shape, we prefer it's unique and not fluid related (now in Lines 142-145).

[Figure]

L147: Please consider replacing "tiny" with smallest

[REPLY] Thank you. We consider the minerals which couldn't be tested with 5μm beam spot as tiny minerals, thus we use tiny rather than smallest, actually, only one particle would be smallest (now in Line 155).

L149: Do you have a record of where the analyses were taken, e.g., cores vs rims of grains?

[REPLY] Thank you. Yeah, as for matrix mineral, we analyzed them along the rimcore-rim transverses (see examples in Figure 5A-D, the red dotted arrows indicate the analyzing direction, the compositional traverse is correspond to that depicted in Figure 9, the test direction is from the left to right). As for prograde and retrograde assemblages, we chose the same horizon with analyzed matrix as much as possible, and then we analyzed them with random directions (now in Lines 156-158).

L155: Please define how you calculate garnet end-members. Is this variation quoted between samples? Is within sample variation in garnet composition negligible?

[REPLY] Thank you. $X_{Alm}$ is defined as $Fe^{2+}/(Ca+Fe^{2+}+Mg+Mn)$, $X_{Prp}$ is $Mg/(Ca+Fe^{2+}+Mg+Mn)$, $X_{Grs}$ is $Ca/(Ca+Fe^{2+}+Mg+Mn)$ and $X_{Sps}$ is $Mn/(Ca+Fe^{2+}+Mg+Mn)$. This variation includes differences both within individual sample (the rim and core) and differences between samples. The diffusion occurred at the very rims, so the variation within one same sample is not negligible; when we do calculations, we chose the composition of plateau cores (now in Lines 211-215).

L156–157: What chemistry would you expect in mantle xenoliths or eclogite? Please be a little more explicit, as it's not necessarily common knowledge amongst all Solid Earth readers.

[REPLY] Thank you. We have explained this in the manuscript (now in Lines 263-264).

L159–161: There looks to be a slight Mn peak in the rim of 17D90 and 17D80 also consistent with a little bit of resorption. Be explicit in terms of the petrographic evidence for resorption.

[REPLY] Thank you. We consider that the petrographic evidence is the existence of "white-eye socket" symplectite, which consumed the boundary between garnet and clinopyroxene, and then caused resorption (now in Lines 265-270).

S1 and S2: Please include recalculated cations in data tables. S1 would be useful as a standard table in the main text, rather than as supplementary information.

[REPLY] Thank you. We have placed Table S1 in the main text (now they are Tables 1-4, now in Line 257), and cations of each mineral composition are calculated and now listed in Tables 1-4, Table S1 and Table S2 (they are all current table numbers).

L164–165: "the chemical composition" Also here, please clarify what you mean by "altered to a different extent." Different to what?

[REPLY] Thank you. What we want to express is that near the internal location of exsolution, chemical composition of clinopyroxene might be changed, it might be related to what kind of the exsolution is and other unknown factors (now in Lines 273-274).

L167–168 and L175: Your calculations are highly dependent on these 'reintegrated' compositions, so please provide some information about the approach taken to calculate these? Did you estimate the proportion of exsolution lamellae and reincorporate their chemistry according to this mode?

[REPLY] Thank you. We have mentioned the method of recalculation in the note of present Table 1 and Table 2 (now in Lines 848-850 and 859-861). We estimated the proportion of exsolution lamellae and reincorporated their chemistry to the host clinopyroxene. We admit that the error of this method is random, because we could only see the planar distribution rather than from three dimensions (now in Lines 213-215 and 276-278).

L168–170: Additional references here could include Carswell et al, Min. Mag (1996) and Castelli, JMG (1991). It should also be noted that these references and those in your current manuscript only document high-Al and F titanite in marbles and calcsilicates,

if your measured Al2O3 reflects high P metamorphism (your concentrations are significantly less than those in Franz & Spear, 1985, I believe), your manuscript would (though I'm admittedly not that familiar with the literature so there may be those I'm not aware of) be the first documentation of such compositions in garnet pyroxenites. Frost et al., Chemical Geology (2001) have a good discussion of the expected metamorphic stability of sphere/titanite in metabasites, and because your results extend their general observations, I recommend expanding your discussion to include this paper.

[REPLY] Thank you. We've added the two references (now in Lines 280-281), and we discussed the stability of titanite within our samples in the discussion part (now in Lines 365-371).

Section: Geothermobarometry: You make some good observations, but it would be useful if there was more evidence for the peak conditions you appeal to, as I think they could be interpreted as having formed during decompression from high-T. I recommend you include the Zr-in-sphene/titanite (Hayden et al, CMP, 2008) thermobarometer which has been shown to sensitive to pressure variations and is calibrated up to 1000 ËŽC with ±20 ËŽC, If coupled with the Zr-in-rutile thermometer and potentially the Ti-in-rutile thermometer (e.g., see Ewing et al., CPM, 2012 and Liu et al., Gondwana Research, 2015) you could provide additional independent constraints on peak P-T. You also have quite a lot of geochemical data, did you consider applying pseudosection thermobarometry (e.g, in Theriak-Domino, Thermocalc or Perplex) or the AveragePT approach in Thermocalc (especially now that there are recently updated models for your chemical system)? For the former, you might get quite large, low variance fields, but it might be a useful exercise to see compare with your more conventional barometers.

[REPLY] Thank you. We have employed these two geothermometers for calculation. However, there is no quartz/coesite or zircon in our samples, therefore we assumed a set of $a_{SiO2}$ for further calculation. We discuss this in detail in Lines 224-254, 428-456.

As for pseudosection thermobarometry, it is known that these bimineralic rocks would occupy large domains in phase diagram, and thus could not obtain accurate *P-T* results. The AveragePT method is also challenging for these bimineralic assemblage rocks because there are insufficient model reactions could be used to constrain the *P-T* conditions.

L174: I don't think it's appropriate to simply use the average chemical composition of garnet with matrix clinopyroxene for conventional thermobarometry to obtain conditions of peak metamorphism unless garnet chemistries are completely homogenous, and you think this homogenization occurred at peak. If you make the assumption that this is the case, it is worth being explicit with this. Your garnets in 17D80 and 17D95 are approximately homogeneous, but rim chemistries might have been incorporated during cooling. You should not use these rim chemistries at all, and only use plateau cores. There is some Mn zoning preserved in the core of 17D78! I would suggest being a little more targeted with the use of these chemistries when using them with conventional thermobarometers.

[REPLY] Thank you. Yes, we assume chemical composition of peak assemblage are completely homogenous at the metamorphic peak. Yes, we actually used the average chemical composition of plateau cores of garnet which represents the peak composition. Because resorption occurred at the rims, so we didn't use these rim chemistries for calculation of average chemical composition (now in Lines 213-215).

L179: I recommend you include the error estimates (±4 kbar) in your Figure 9.

[REPLY] Thank you. We have included the error estimates (±4 kbar) (now in Figure 10).

L179–180: "...assemblages mainly consist of plagioclase..."

[REPLY] Thank you. We have rephrased this sentence (now in Lines 219-220).

L182: Please include errors with hornblende thermobarometers!

[REPLY] Thank you. We have added the errors with hornblende thermobarometers (now in Lines 220-222).

L192: I don't think you've necessarily "certified UHP metamorphism." Generally, this is only really truly the case if coesite pseudomorphs or marjoritic garnet is documented. While your data might well suggest this as it currently stands, your results are highly dependent on results of a single barometer which uses chemistries that are highly modified to their present state. I also find it strange that your jadeite component in clinopyroxene is so low in the chemistry you infer as having equilibrated at UHP conditions; why might this be?

[REPLY] Thank you. No coesite pseudomorph or marjoritic garnet was found in our samples, that's true. That is because of the low $SiO_2$ component of these rocks, i.e. their chemistry is almost ultrabasic. No coesite or marjoritic garnet can be formed in these rocks even if they experienced UHP metamorphism. As people have known that, mineral assemblages and mineral chemical compositions in metamorphic rocks depend not only on P-T conditions, but also on bulk composition. For example, pure marble never grows coesite or garnet even if it reached UHP conditions. We tested the bulk-rock compositions (now in Table S3) and found it's sodium deficient, this could explain why jadeite component in clinopyroxene is too low. Therefore, we represent other petrographic evidences (Figure 6) and geothermobarometry constrains to demonstrate UHP metamorphism.

L193–194: Exsolution of rutile lamellae in garnet is not exclusive to decompression from UHP metamorphism (as far as I am aware). It can also be documented as having formed during the retrogression of mafic garnet granulites from high-T conditions so is

not necessarily diagnostic.

[REPLY] Thank you. Yes, as demonstrated by Ague and Eckert (2012) and others, exsolution of rutile lamellae in garnet usually represents extremely high P or high T conditions, so we combined with the occurrence of high-Al titanite as petrographic evidences to draw the conclusion of UHP metamorphism, rather than HT metamorphism (see detailed in discussion, now in Lines 372-381).

L203: Remove "scenario"

[REPLY] Thank you. We have rephrased this sentence (now in Lines 185-186).

Dating metamorphism and L226: Please include geochronological data tables in the main text, rather than as supplementary tables! Are the ages you quote weighted averages? Also, in Supplementary Table 4 you quote uncertainties as "±s" which is a good convention to follow; are the ± intervals quoted in the text also s, rather than sigma?

[REPLY] Thank you. We have placed the geochronological data table in the main text (now it's Table 8). The final U-Pb age result is weighted average of data in Table S4 (now it's Table 8), and quoted with 95% confidence level, but the ages in Table S4 (now it's Table 8) are individual analytical results from lab directly (now in Lines 208-209). The "±s" should be "±σ", and we have changed this in present Table 8.

L221: Please correct to "Terra-Wasserburg" Nice ages!!

[REPLY] Thank you. Sorry, but we don't understand why this should be correct to "Terra-Wasserburg", the authors' name is Tera and Wasserburg (now in Line 204).

L221–224: That's ok, but if titanite formed at the expense of a high-μ phase (e.g., rutile) then it will most likely incorporate more radiogenic Pb than the bulk rock, potentially

causing problems with your common Pb projection. Any thoughts on what reactions may have formed titanite?

[REPLY] Thank you. Titanite is mainly composed of $CaTiSiO_5$, thus the source of Ti is very important. Rutile, clinopyroxene and sometimes even garnet all could account for the Ti source. Our observation suggests that rutile and clinopyroxene could form titanite during retrograde. If some of them were formed at the metamorphic peak, they might be formed at the expense of hornblende, but we didn't see any such reaction textures. Because titanite was picked out and then analyzed, we couldn't exclude the possibility of what you said (the analyzed titanite might formed at the expense of rutile) (now in Lines 204-207).

L227Ăn–232: The titanites in 17D90 and 17D95 are altered? Do you mean by diffusion, or something else like fluid fluxing? The zoning in the BSE looks to me to be primary, perhaps? In places, it's concentric, and the sharp boundaries are consistent with negligible diffusional homogenization. Did you consider trying to put more multiple spots down on individual grains? In a grain in 17D95 you've done this, and the core age gives a nearly 20 Myr younger age than the rim, which is interesting/strange and potentially worth investigating! Patchy zoning such as this has been related to interface coupled dissolution and precipitation reactions (Bonamici et al, 2015; Garber et al, 2015; Walters Kohn, 2017), so your observations might provide a nice framework. See next comment for more on this.

[REPLY] Thank you. Yes, we think they might be altered due to the scattered ages and textural variations (now in Lines 304-310). When we selected spots on titanite, we not only based on BSE images, but also based on transmitted light images and reflected light images to avoid uncertainties which might affect the dating results. In addition, if two spots are too close, the quality of analytical results might be affected, thus we selected two spots on individual grain. The core age gives a nearly 20 Myr younger age than the rim, this might be caused by localized alteration and resetting of U-Pb age (e.g.,

Holder and Hacker, 2019), possibly because the rim is not closed in most sections.

L234–236: While you're correct that titanite U-Pb ages are traditionally thought to record cooling through 600Ăˇ n–750 ËŽC (additional references to those you've provided include Cherniak (1993) and Spear and Parrish, (1996)), recent studies (including experimental studies) have indicated that Pb diffusion seems to be negligible in titanite at T < 850 ËŽC. Some relevant examples include Castelli and Rubatto, CMP, 2002; Garber et al, J Pet, 2017; Walters and Kohn, JMG, 2017; Holder and Hacker, Chemical Geology, 2019. Rather than cooling, these authors document U–Pb titanite dates that record the timing of neo-crystallization, fluid-driven alteration/resetting, or deformation that might facilitate resetting of U-Pb dates and compositions. It is worth considering potential alternative scenarios given increasing evidence that dates such as those you've obtained in this study may not reflect cooling, but some other process.

[REPLY] Thank you. Just like what you said, U-Pb titanite dates shall record the timing of neo-crystallization, or fluid-driven alteration/resetting, or deformation that might facilitate resetting of U-Pb dates and compositions. We think our views don't conflict with each other, we supposed the dating record cooling / retrograde metamorphism, which might cause neo-crystallization or U-Pb resetting. In fact, it's consistent with petrography, as Al-rich titanite should be the peak assemblage, while the dated titanite (not those Al-rich) might be the retrograde assemblages (neo-crystallization) or altered (from Al-rich titanite) (now in Lines 457-469).

L242–243: Maybe replace the end of this sentence with "assess the likelihood of UHP metamorphism"?

[REPLY] Thank you. We have rephrased this sentence (now in Lines 316-317).

L255–258: I'm still not entirely sure why "the prograde assemblage (M1) clearly indicates the subduction process". Please expand on this. "Very deep but different

depths" is not clear, either. Please clarify.

[REPLY] Thank you. It is known that UHP metamorphic rocks can only be formed either underwent very deep subduction or those came from the upper mantle (e.g., mantle xenolith). The former must experience prograde metamorphism (from shallow level, low P-T conditions to very deep interior of the lithosphere, high P-T conditions); as for the latter, no prograde metamorphism occurred at all. In fact, contact metamorphism, deformation, collision and subduction process can all form prograde assemblages, but apparently, in our work it's the last case. It's temperature is much higher than the granitic magma/pluton and there is no contact metamorphic aureole formed at all. No foliation in garnet clinopyroxene was found. "Very deep but different depths" is referred from the obtained very high pressures and large differences in peak pressures of these rocks (now in Lines 342 and 350-352).

L245–250: This paragraph should be placed in a little more context to account for its inclusion.

[REPLY] Thank you. We have explained the necessity of discussing the rock type firstly in the rewritten paragraph (now in Lines 319-327).

L274–278: I recommend you rephrase this paragraph, but yes, I think the sample could alternatively reflect decompression from high-T conditions (as mentioned in a previous comment), based at least on the microstructures. If there are other barometers that you could apply to this dataset to evidence your UHP conclusion, great! I think the possibility that your pyroxene compositions are not reflective of UHP conditions and instead reflect HT metamorphism should be explored a little more. Would this interpretation potentially fit with the garnet clinopyroxenite field setting (in the undeformed granitic bodies) a little better, too?

[REPLY] Thank you. We don't think the granitic body could be account for

metamorphism of these garnet clinopyroxene even if it's charnockite. We have explained the reasons why our clinopyroxene compositions are poor in jadeitic components in the previous reply. In this paragraph, we combined Al-rich titanite and exsolved rutile lamellae in garnet to suggest UHP metamorphism based on previous knowledge. Thus, our discuss is possibly logical and valid (now in Lines 372-381).

L280–282: What about the potentially important role of intracrystalline diffusional exchange (e.g., Pattison Begin, 19094) that results in Fe-Mg exchange at the relatively high temperatures you suggest? Geothermobarometers that use net transfer reactions and Ca, Al and Si tend to be a little more reliable at these conditions.

[REPLY] Thank you. As demonstrated by Pattison and Begin (1994), intergranular diffusion was fast relative to intragranular diffusion at higher temperatures, which might cause the rock to record lower temperatures than the true peak temperatures. For this reason, we analyzed the largest garnet grains and prepared the compositional maps to obtain the most refractory compositions, and then applied the Grt-Cpx geothermometer of Nakamura (2009). As for geobarometer, we used the Grt-Cpx geobarometer calibrated by Beyer et al (2015), which was based on net-transfer model reactions (now in Lines 383-385).

L293: By "wide representative," do you mean its applicable to a wide range of bulk rock compositions?

[REPLY] Thank you. This thermometer was calibrated for in mafic and ultramafic systems (not suitable for metapelitic or felsic rocks), and it includes wide chemical ranges of garnet and clinopyroxene, which is consistent with our samples (now in Line 396).

L315: Here and in a previous paragraph, you justify use of the geothermobaromers as your chosen calibrations are "most accurate." I would recommend modifying this

discussion, because results are only accurate if you know the true result, how do you know the calibration is accurate for your bulk composition? Has it been calibrated for such (in which case, great!!).

[REPLY] Thank you. What we mean is that among all the geothermobarometers, we chose the most suitable calibrations to do calculation for our samples (now in Line 417).

L335–338: A subduction mélange? Also recommend changing "...large gaps of the peak..." to "...large differences in the peak..."

[REPLY] Thank you. Yes, we think these rock assemblages show features of a mélange. We have rephrased this sentence (now in Lines 475-478).

L345: consider rephrasing "which is neglected to different extent in the past"

[REPLY] Thank you. We have deleted this sentence.

**References**

Ague, J. J., and Eckert, J. O.: Precipitation of rutile and ilmenite needles in garnet: Implications for extreme metamorphic conditions in the Acadian Orogen, U.S.A, Am. Mineral., 97, 840-855, https://doi.org/10.2138/am.2012.4015, 2012.

Beyer, C., Frost, D. J., and Miyajima, N.: Experimental calibration of a garnet–clinopyroxene geobarometer for mantle eclogites, Contrib. Mineral. Petr., 169, 1-21, https://doi.org/10.1007/s00410-015-1113-z, 2015.

Cunningham, D., Zhang, J., and Li, Y.: Late Cenozoic transpressional mountain building directly north of the Altyn Tagh Fault in the Sanweishan and Nanjieshan, North Tibetan Foreland, China, Tectonophysics, 687, 111-128, http://doi.org/10.1016/j.tecto.2016.09.010, 2016.

Garber, J.M., Hacker, B.R., Kylander-Clark, A.R.C., Stearns, M., and Seward, G.: Controls on Trace Element Uptake in Metamorphic Titanite: Implications for

Petrochronology, J. Petrol., 58, 1031-1058, https://doi.org/10.1093/petrology/egx046, 2017

Hayden, L.A., Watson, E.B., and Wark, D.A.: A thermobarometer for sphene (titanite), Contrib. Mineral. Petr., 155, 529-540, https://doi.org/10.1007/s00410-007-0256-y, 2008.

Holder, R.M., and Hacker, B.R.: Fluid-driven resetting of titanite following ultrahigh-temperature metamorphism in southern Madagascar, Chem. Geol., 504, 38-52, https://doi.org/10.1016/j.chemgeo.2018.11.017, 2019.

Morimoto, N.: Nomenclature of Pyroxenes, Mineral. Petrol., 39, 55–76, https://doi.org/10.1007/BF01226262, 1988.

Nakamura, D.: A new formulation of garnet–clinopyroxene geothermometer based on accumulation and statistical analysis of a large experimental data set, J. Metamorph. Geol., 27, 495-508, https://doi.org/10.1111/j.1525-1314.2009.00828.x, 2009.

Pattison, D.R.M., and Begin, N.J.: Zoning patterns in orthopyroxene and garnet in granulites: implications for geothermometry, J. Metamorph. Geol., 12, 387-410, https://doi.org/10.1111/j.1525-1314.1994.tb00031.x, 1994.

Wang, H. Y. C., Chen, H.-X., Zhang, Q. W. L., Shi, M.-Y., Yan, Q.-R., Hou, Q.-L., Zhang, Q., Kusky, T., and Wu, C.-M.: Tectonic mélange records the Silurian–Devonian subduction-metamorphic process of the southern Dunhuang terrane, southernmost Central Asian Orogenic Belt, Geology, 45, 427-430, https://doi.org/10.1130/g38834.1, 2017a.

Wang, H. Y. C., Zhang, Q. W. L., Lu, J.-S., Chen, H.-X., Liu, J.-H., Zhang, H. C. G., Pham, V. T., Peng, T., and Wu, C.-M.: Metamorphic evolution and geochronology of the tectonic mélange of the Dongbatu and Mogutai blocks, middle Dunhuang orogenic belt, northwestern China, Geosphere, 14, 883-906, https://doi.org/10.1130/ges01514.1, 2018b.

Ye, K., Cong, B., and Ye, D.: The possible subduction of continental material to depths greater than 200km, Nature, 407, 734-736, https://doi.org/10.1038/35037566, 2000.

You, Z.D., Zhong, Z.Q., and Suo, S.T.: The mineralogical criteria for ultra-high pressure metamorphism, Geoscience, 21, 195-202, https://doi.org/10.3969/j.issn.1000-8527.2007.02.003, 2007. [in Chinese with English abstract]

Zhang, Q. W. L., Wang, H. Y. C., Liu, J.-H., Shi, M.-Y., Chen, Y.-C., Li, Z. M. G., and Wu, C.-M.: Diverse subduction and exhumation of tectono-metamorphic slices in the Kalatashitage area, western Paleozoic Dunhuang Orogenic Belt, northwestern China, Lithos, 360-361, https://doi.org/10.1016/j.lithos.2020.105434, 2020.

Zhang, H.C.G., Liu, J.H., Chen, Y.C., Zhang, Q.W.L., Pham, V.T., Peng, T., Li, Z.M.G., and Wu, C.M.: Neoarchean metamorphic evolution and geochronology of the Miyun metamorphic complex, North China Craton, Precambrian. Res., 320, 78-92, https://doi.org/10.1016/j.precamres.2018.10.015, 2019.

---

## Author Comment (AC2) · 7 Sep 2020

The replies are immediately listed below the comment
Ultra-high pressure (UHP) metamorphic rocks of continental affinity indicate that continental slabs can subduct to great depths where coesite and diamond can stabilize, and UHP metamorphism has been a hot topic for the past decades. Li et al. (2020) for the first time report UHP garnet clinopyroxenites from the Paleozoic Dunhuang orogenic belt, NW China. The rocks are retrieved to show clockwise P–T paths with the peak conditions of 790âĹij920°C / 28âĹij41 kbar that are constrained by available garnet-clinopyroxene thermobarometries. This UHP metamorphism can be further confirmed by the presence of high-Al titanite inclusions in garnet and pyrope-rich garnet with exsolved rutile lamellae. Titanite SIMS U-Pb dating yields a metamorphic age of 389âĹij370 Ma, interpreted to represent the post peak exhumation time. The evidence for the UHP metamorphism is robust and the age data are in good quality. It will be much better to involve available bulk-rock compositions for both major and trace elements because these are significant for understand the petrogenetic origin of the UHP rocks. The discovery of the UHP garnet clinopyroxenites is great advance for the study of the Dunhuang orogenic belt, which has added a new case for the globe occurrences of UHP metamorphic rocks.

[REPLY] Thanks for your suggestions. We have presented bulk-rock compositions for both major and trace elements in supplementary Table S3 and discussed petrogenesis of garnet clinopyroxenite enclaves in section "Protoliths of garnet clinopyroxenite" of discussion part (now in Lines 328-344 and 349-352). From discrimination diagrams, the protoliths of these rocks are basalt. But, $SiO_2$ contents of the rocks are between 42.2 to 47.7 wt%. Thus, we inferred that the bulk rock compositions had possibly been modified in the metamorphism or alternatively, the rocks experienced chemical weathering prior to metamorphism.

Comments from Chunjing Wei at the School of Earth and Space Sciences, Peking University, China. cjwei@pku.edu.cn Interactive comment on Solid Earth Discuss., https://doi.org/10.5194/se-2020-95, 2020.